# Towards Understanding Regularization in Batch Normalization

**Ping Luo**[1,3*]   **Xinjiang Wang**[2*]   **Wenqi Shao**[1*]   **Zhanglin Peng**[2]
[1]The Chinese University of Hong Kong   [2]SenseTime Research   [3]The University of Hong Kong

## Abstract

Batch Normalization (BN) improves both convergence and generalization in training neural networks. This work understands these phenomena theoretically. We analyze BN by using a basic block of neural networks, consisting of a kernel layer, a BN layer, and a nonlinear activation function. This basic network helps us understand the impacts of BN in three aspects. First, by viewing BN as an implicit regularizer, BN can be decomposed into population normalization (PN) and gamma decay as an explicit regularization. Second, learning dynamics of BN and the regularization show that training converged with large maximum and effective learning rate. Third, generalization of BN is explored by using statistical mechanics. Experiments demonstrate that BN in convolutional neural networks share the same traits of regularization as the above analyses.

## 1  Introduction

Batch Normalization (BN) is an indispensable component in many deep neural networks (He et al., 2016; Huang et al., 2017). BN has been widely used in various areas such as machine vision, speech and natural language processing. Experimental studies (Ioffe & Szegedy, 2015) suggested that BN improves convergence and generalization by enabling large learning rate and preventing overfitting when training deep networks. Understanding BN theoretically is a key question.

This work investigates regularization of BN as well as its optimization and generalization in a single-layer perceptron, which is a building block of deep models, consisting of a kernel layer, a BN layer, and a nonlinear activation function such as ReLU. The computation of BN is written by

$$y = g(\hat{h}), \ \ \hat{h} = \gamma \frac{h - \mu_\mathcal{B}}{\sigma_\mathcal{B}} + \beta \ \text{ and } \ h = \mathbf{w}^\mathsf{T}\mathbf{x}. \tag{1}$$

This work denotes a scalar and a vector by using lowercase letter (*e.g.* $x$) and bold lowercase letter (*e.g.* $\mathbf{x}$) respectively. In Eqn.(1), $y$ is the output of a neuron, $g(\cdot)$ denotes an activation function, $h$ and $\hat{h}$ are hidden values before and after batch normalization, $\mathbf{w}$ and $\mathbf{x}$ are kernel weight vector and network input respectively. In BN, $\mu_\mathcal{B}$ and $\sigma_\mathcal{B}$ represent the mean and standard deviation of $h$. They are estimated within a batch of samples for each neuron independently. $\gamma$ is a scale parameter and $\beta$ is a shift parameter. In what follows, Sec.1.1 overviews assumptions and main results, and Sec.1.2 presents relationships with previous work.

### 1.1  Overview of Results

We overview results in three aspects.

• First, Sec.2 decomposes BN into population normalization (PN) and gamma decay. To better understand BN, we treat a single-layer perceptron with ReLU activation function as an illustrative case. Despite the simplicity of this case, it is a building block of deep networks and has been widely adopted in theoretical analyses such as proper initialization (Krogh & Hertz, 1992; Advani & Saxe, 2017), dropout (Wager et al., 2013), weight decay and data augmentation (Bös, 1998). The results in Sec.2 can be extended to deep neural networks as presented in Appendix C.4.

Our analyses assume that neurons at the BN layer are independent similar to (Salimans & Kingma, 2016; van Laarhoven, 2017; Teye et al., 2018), as the mean and the variance of BN are estimated individually for each neuron of each layer. The form of regularization in this study does not rely on

---

*The first three authors contribute equally. Corresponding to pluo.lhi@gmail.com, {wangxinjiang, pengzhanglin}@sensetime.com, weqish@link.cuhk.edu.hk.

Gaussian assumption on the network input and the weight vector, meaning our assumption is milder than those in (Yoshida et al., 2017; Ba et al., 2016; Salimans & Kingma, 2016).

Sec.2 tells us that BN has an explicit regularization form, gamma decay, where $\mu_\mathcal{B}$ and $\sigma_\mathcal{B}$ have different impacts: (1) $\mu_\mathcal{B}$ discourages reliance on a single neuron and encourages different neurons to have equal magnitude, in the sense that corrupting individual neuron does not harm generalization. This phenomenon was also found empirically in a recent work (Morcos et al., 2018), but has not been established analytically. (2) $\sigma_\mathcal{B}$ reduces kurtosis of the input distribution as well as correlations between neurons. (3) The regularization strengths of these statistics are inversely proportional to the batch size $M$, indicating that BN with large batch would decrease generalization. (4) Removing either one of $\mu_\mathcal{B}$ and $\sigma_\mathcal{B}$ could imped convergence and generalization.

• Second, by using ordinary differential equations (ODEs), Sec.3 shows that gamma decay enables the network trained with BN to converge with large maximum learning rate and effective learning rate, compared to the network trained without BN or trained with weight normalization (WN) (Salimans & Kingma, 2016) that is a counterpart of BN. The maximum learning rate (LR) represents the largest LR value that allows training to converge to a fixed point without diverging, while effective LR represents the actual LR in training. Larger maximum and effective LRs imply faster convergence rate.

• Third, Sec.4 compares generalization errors of BN, WN, and vanilla SGD by using statistical mechanics. The "large-scale" regime is of interest, where number of samples $P$ and number of neurons $N$ are both large but their ratio $P/N$ is finite. In this regime, the generalization errors are quantified both analytically and empirically.

Numerical results in Sec.5 show that BN in CNNs has the same traits of regularization as disclosed above.

## 1.2 RELATED WORK

**Neural Network Analysis**. Many studies conducted theoretical analyses of neural networks (Opper et al., 1990; Saad & Solla, 1996; Bs & Opper, 1998; Pennington & Bahri, 2017; Zhang et al., 2017b; Brutzkus & Globerson, 2017; Raghu et al., 2017; Mei et al., 2016; Tian, 2017). For example, for a multilayer network with linear activation function, Glorot & Bengio (2010) explored its SGD dynamics and Kawaguchi (2016) showed that every local minimum is global. Tian (2017) studied the critical points and convergence behaviors of a 2-layered network with ReLU units. Zhang et al. (2017b) investigated a teacher-student model when the activation function is harmonic. In (Saad & Solla, 1996), the learning dynamics of a committee machine were discussed when the activation function is error function $\mathrm{erf}(x)$. Unlike previous work, this work analyzes regularization emerged in BN and its impact to both learning and generalization, which are still unseen in the literature.

**Normalization**. Many normalization methods have been proposed recently. For example, BN (Ioffe & Szegedy, 2015) was introduced to stabilize the distribution of input data of each hidden layer. Weight normalization (WN) (Salimans & Kingma, 2016) decouples the lengths of the network parameter vectors from their directions, by normalizing the parameter vectors to unit length. The dynamic of WN was studied by using a single-layer network (Yoshida et al., 2017). Li et al. (2018) diagnosed the compatibility of BN and dropout (Srivastava et al., 2014) by reducing the variance shift produced by them.

Moreover, van Laarhoven (2017) showed that weight decay has no regularization effect when using together with BN or WN. Ba et al. (2016) demonstrated when BN or WN is employed, back-propagating gradients through a hidden layer is scale-invariant with respect to the network parameters. Santurkar et al. (2018) gave another perspective of the role of BN during training instead of reducing the covariant shift. They argued that BN results in a smoother optimization landscape and the Lipschitzness is strengthened in networks trained with BN. However, both analytical and empirical results of regularization in BN are still desirable. Our study explores regularization, optimization, and generalization of BN in the scenario of online learning.

**Regularization**. Ioffe & Szegedy (2015) conjectured that BN implicitly regularizes training to prevent overfitting. Zhang et al. (2017a) categorized BN as an implicit regularizer from experimental results. Szegedy et al. (2015) also conjectured that in the Inception network, BN behaves similar to dropout to improve the generalization ability. Gitman & Ginsburg (2017) experimentally compared

BN and WN, and also confirmed the better generalization of BN. In the literature there are also implicit regularization schemes other than BN. For instance, random noise in the input layer for data augmentation has long been discovered equivalent to a weight decay method, in the sense that the inverse of the signal-to-noise ratio acts as the decay factor (Krogh & Hertz, 1992; Rifai et al., 2011). Dropout (Srivastava et al., 2014) was also proved able to regularize training by using the generalized linear model (Wager et al., 2013).

## 2 A PROBABILISTIC INTERPRETATION OF BN

The notations in this work are summarized in Appendix Table 2 for reference.

Training the above single-layer perceptron with BN in Eqn.(1) typically involves minimizing a negative log likelihood function with respect to a set of network parameters $\theta = \{\mathbf{w}, \gamma, \beta\}$. Then the loss function is defined by

$$\frac{1}{P} \sum_{j=1}^{P} \ell(\hat{h}^j) = -\frac{1}{P} \sum_{j=1}^{P} \log p(y^j | \hat{h}^j; \theta) + \zeta \|\theta\|_2^2, \tag{2}$$

where $p(y^j | \hat{h}^j; \theta)$ represents the likelihood function of the network and $P$ is number of training samples. As Gaussian distribution is often employed as prior distribution for the network parameters, we have a regularization term $\zeta \|\theta\|_2^2$ known as weight decay (Krizhevsky et al., 2012) that is a popular technique in deep learning, where $\zeta$ is a coefficient.

To derive regularization of BN, we treat $\mu_{\mathcal{B}}$ and $\sigma_{\mathcal{B}}$ as random variables. Since one sample $\mathbf{x}$ is seen many times in the entire training course, and at each time $\mathbf{x}$ is presented with the other samples in a batch that is drawn randomly, $\mu_{\mathcal{B}}$ and $\sigma_{\mathcal{B}}$ can be treated as injected random noise for $\mathbf{x}$.

**Prior of $\mu_{\mathcal{B}}, \sigma_{\mathcal{B}}$.** By following (Teye et al., 2018), we find that BN also induces Gaussian priors for $\mu_{\mathcal{B}}$ and $\sigma_{\mathcal{B}}$. We have $\mu_{\mathcal{B}} \sim \mathcal{N}(\mu_{\mathcal{P}}, \frac{\sigma_{\mathcal{P}}^2}{M})$ and $\sigma_{\mathcal{B}} \sim \mathcal{N}(\sigma_P, \frac{\rho+2}{4M})$, where $M$ is batch size, $\mu_{\mathcal{P}}$ and $\sigma_{\mathcal{P}}$ are population mean and standard deviation respectively, and $\rho$ is kurtosis that measures the peakedness of the distribution of $h$. These priors tell us that $\mu_{\mathcal{B}}$ and $\sigma_{\mathcal{B}}$ would produce Gaussian noise in training. There is a tradeoff regarding this noise. For example, when $M$ is small, training could diverge because the noise is large. This is supported by experiment of BN (Wu & He, 2018) where training diverges when $M = 2$ in ImageNet (Russakovsky et al., 2015). When $M$ is large, the noise is small because $\mu_{\mathcal{B}}$ and $\sigma_{\mathcal{B}}$ get close to $\mu_{\mathcal{P}}$ and $\sigma_{\mathcal{P}}$. It is known that $M > 30$ would provide a moderate noise, as the sample statistics converges in probability to the population statistics by the weak Law of Large Numbers. This is also supported by experiment (Ioffe & Szegedy, 2015) where BN with $M = 32$ already works well in ImageNet.

### 2.1 A REGULARIZATION FORM

The loss function in Eqn.(2) can be written as an expected loss by integrating over the priors of $\mu_{\mathcal{B}}$ and $\sigma_{\mathcal{B}}$, that is, $\frac{1}{P} \sum_{j=1}^{P} \mathbb{E}_{\mu_{\mathcal{B}}, \sigma_{\mathcal{B}}}[\ell(\hat{h}^j)]$ where $\mathbb{E}[\cdot]$ denotes expectation. We show that $\mu_{\mathcal{B}}$ and $\sigma_{\mathcal{B}}$ impose regularization on the scale parameter $\gamma$ by decomposing BN into population normalization (PN) and gamma decay. To see this, we employ a single-layer perceptron and ReLU activation function as an illustrative example. A more rigorous description is provided in Appendix C.1.

**Regularization of $\mu_{\mathcal{B}}, \sigma_{\mathcal{B}}$.** Let $\ell(\hat{h})$ be the loss function defined in Eqn.(2) and ReLU be the activation function. We have

$$\frac{1}{P} \sum_{j=1}^{P} \mathbb{E}_{\mu_{\mathcal{B}}, \sigma_{\mathcal{B}}} \ell(\hat{h}^j) \simeq \underbrace{\frac{1}{P} \sum_{j=1}^{P} \ell(\bar{h}^j)}_{\text{PN}} + \underbrace{\zeta(h)\gamma^2}_{\text{gamma decay}}, \quad \text{and} \quad \zeta(h) = \underbrace{\frac{\rho+2}{8M} \mathcal{I}(\gamma)}_{\text{from } \sigma_{\mathcal{B}}} + \underbrace{\frac{1}{2M} \frac{1}{P} \sum_{j=1}^{P} \sigma(\bar{h}^j)}_{\text{from } \mu_{\mathcal{B}}}, \tag{3}$$

where $\bar{h}^j = \gamma \frac{h^j - \mu_{\mathcal{P}}}{\sigma_{\mathcal{P}}} + \beta$ and $h^j = \mathbf{w}^\mathsf{T} \mathbf{x}^j$ represent the computations of PN. $\zeta(h)\gamma^2$ represents gamma decay, where $\zeta(h)$ is an adaptive decay factor depended on the hidden value $h$. Moreover, $\rho$ is the kurtosis of distribution of $h$, $\mathcal{I}(\gamma)$ represents an estimation of the Fisher information of $\gamma$ and $\mathcal{I}(\gamma) = \frac{1}{P} \sum_{j=1}^{P} (\frac{\partial \ell(\hat{h}^j)}{\partial \gamma})^2$, and $\sigma(\cdot)$ is a sigmoid function.

From Eqn.(3), we have several observations that have both theoretical and practical values.

• First, PN replaces the batch statistics $\mu_{\mathcal{B}}, \sigma_{\mathcal{B}}$ in BN by the population statistics $\mu_{\mathcal{P}}, \sigma_{\mathcal{P}}$. In gamma decay, computation of $\zeta(h)$ is data-dependent, making it differed from weight decay where the

coefficient is determined manually. In fact, Eqn.(3) recasts the randomness of BN in a deterministic manner, not only enabling us to apply methodologies such as ODEs and statistical mechanics to analyze BN, but also inspiring us to imitate BN's performance by WN without computing batch statistics in empirical study.

● Second, PN is closely connected to WN, which is independent from sample mean and variance. WN (Salimans & Kingma, 2016) is defined by $\upsilon \frac{\mathbf{w}^T \mathbf{x}}{||\mathbf{w}||_2}$ that normalizes the weight vector $\mathbf{w}$ to have unit variance, where $\upsilon$ is a learnable parameter. Let each diagonal element of the covariance matrix of $\mathbf{x}$ be $a$ and all the off-diagonal elements be zeros. $\bar{h}^j$ in Eqn.(3) can be rewritten as

$$\bar{h}^j = \gamma \frac{\mathbf{w}^T \mathbf{x}^j - \mu_\mathcal{P}}{\sigma_\mathcal{P}} + \beta = \upsilon \frac{\mathbf{w}^T \mathbf{x}^j}{||\mathbf{w}||_2} + b, \tag{4}$$

where $\upsilon = \frac{\gamma}{a}$ and $b = -\frac{\gamma \mu_\mathcal{P}}{a||\mathbf{w}||_2} + \beta$. Eqn.(4) removes the estimations of statistics and eases our analyses of regularization for BN.

● Third, $\mu_\mathcal{B}$ and $\sigma_\mathcal{B}$ produce different strengths in $\zeta(h)$. As shown in Eqn.(3), the strength from $\mu_\mathcal{B}$ depends on the expectation of $\sigma(\bar{h}^j) \in [0, 1]$, which represents excitation or inhibition of a neuron, meaning that a neuron with larger output may exposure to larger regularization, encouraging different neurons to have equal magnitude. This is consistent with empirical result (Morcos et al., 2018) which prevented reliance on single neuron to improve generalization. The strength from $\sigma_\mathcal{B}$ works as a complement for $\mu_\mathcal{B}$. For a single neuron, $\mathcal{I}(\gamma)$ represents the norm of gradient, implying that BN punishes large gradient norm. For multiple neurons, $\mathcal{I}(\gamma)$ is the Fisher information matrix of $\gamma$, meaning that BN would penalize correlations among neurons. Both $\sigma_\mathcal{B}$ and $\mu_\mathcal{B}$ are important, removing either one of them would imped performance.

**Extensions to Deep Networks.** The above results can be extended to deep networks as shown in Appendix C.4 by decomposing the expected loss at a certain hidden layer. We also demonstrate the results empirically in Sec.5, where we observe that CNNs trained with BN share similar traits of regularization as discussed above.

## 3 Optimization with Regularization

Now we show that BN converges with large maximum and effective learning rate (LR), where the former one is the largest LR when training converged, while the latter one is the actual LR during training. With BN, we find that both LRs would be larger than a network trained without BN. Our result explains why BN enables large learning rates used in practice (Ioffe & Szegedy, 2015).

Our analyses are conducted in three stages. First, we establish dynamical equations of a teacher-student model in the thermodynamic limit and acquire the fixed point. Second, we investigate the eigenvalues of the corresponding Jacobian matrix at this fixed point. Finally, we calculate the maximum and the effective LR.

**Teacher-Student Model**. We first introduce useful techniques from statistical mechanics (SM). With SM, a student network is dedicated to learn relationship between a Gaussian input and an output by using a weight vector $\mathbf{w}$ as parameters. It is useful to characterize behavior of the student by using a teacher network with $\mathbf{w}^*$ as a ground-truth parameter vector. We treat single-layer perceptron as a student, which is optimized by minimizing the euclidian distance between its output and the supervision provided by a teacher without BN. The student and the teacher have identical activation function.

**Loss Function.** We define a loss function of the above teacher-student model by $\frac{1}{P} \sum_{j=1}^{P} \ell(\mathbf{x}^j) = \frac{1}{P} \sum_{j=1}^{P} \left[ g(\mathbf{w}^{*\mathsf{T}} \mathbf{x}^j) - g(\sqrt{N} \gamma \frac{\mathbf{w}^\mathsf{T} \mathbf{x}^j}{||\mathbf{w}||_2}) \right]^2 + \zeta \gamma^2$, where $g(\mathbf{w}^{*\mathsf{T}} \mathbf{x}^j)$ represents supervision from the teacher, while $g(\sqrt{N} \gamma \frac{\mathbf{w}^\mathsf{T} \mathbf{x}^j}{||\mathbf{w}||_2})$ is the output of student trained to mimic the teacher. This student is defined by following Eqn.(4) with $\nu = \sqrt{N} \gamma$ and the bias term is absorbed into $\mathbf{w}$. The above loss function represents BN by using WN with gamma decay, and it is sufficient to study the learning rates of different approaches. Let $\theta = \{\mathbf{w}, \gamma\}$ be a set of parameters updated by SGD, *i.e.* $\theta^{j+1} = \theta^j - \eta \frac{\partial \ell(\mathbf{x}^j)}{\partial \theta^j}$ where $\eta$ denotes learning rate. The update rules for $\mathbf{w}$ and $\gamma$ are

$$\mathbf{w}^{j+1} - \mathbf{w}^j = \eta \delta^j \left( \frac{\gamma^j \sqrt{N}}{||\mathbf{w}^j||_2} \mathbf{x}^j - \frac{\tilde{\mathbf{w}}^{j\mathsf{T}} \mathbf{x}^j}{||\mathbf{w}^j||_2^2} \mathbf{w}^j \right) \text{ and } \gamma^{j+1} - \gamma^j = \eta \left( \frac{\delta^j \sqrt{N} \mathbf{w}^{j\mathsf{T}} \mathbf{x}^j}{||\mathbf{w}^j||_2} - \zeta \gamma^j \right), \tag{5}$$

where $\tilde{\mathbf{w}}^j$ denotes a normalized weight vector of the student, that is, $\tilde{\mathbf{w}}^j = \sqrt{N}\gamma^j \frac{\mathbf{w}^j}{\|\mathbf{w}^j\|_2}$, and $\delta^j = g'(\tilde{\mathbf{w}}^{j\mathsf{T}}\mathbf{x}^j)[g(\mathbf{w}^{*\mathsf{T}}\mathbf{x}^j) - g(\tilde{\mathbf{w}}^{j\mathsf{T}}\mathbf{x}^j)]$ represents the gradient[1] for clarity of notations.

**Order Parameters**. As we are interested in the "large-scale" regime where both $N$ and $P$ are large and their ratio $P/N$ is finite, it is difficult to examine a student with parameters in high dimensions directly. Therefore, we transform the weight vectors to order parameters that fully characterize interactions between the student and the teacher network. In this case, the parameter vector can be reparameterized by using a vector of three elements including $\gamma$, $R$, and $L$. In particular, $\gamma$ measures length of the normalized weight vector $\tilde{\mathbf{w}}$, that is, $\tilde{\mathbf{w}}^{\mathsf{T}}\tilde{\mathbf{w}} = N\gamma^2 \frac{\mathbf{w}^{\mathsf{T}}\mathbf{w}}{\|\mathbf{w}\|_2^2} = N\gamma^2$. The parameter $R$ measures angle (overlapping ratio) between the weight vectors of student and teacher. We have $R = \frac{\tilde{\mathbf{w}}^{\mathsf{T}}\mathbf{w}^*}{\|\tilde{\mathbf{w}}\|\|\mathbf{w}^*\|} = \frac{1}{N\gamma}\tilde{\mathbf{w}}^{\mathsf{T}}\mathbf{w}^*$, where the norm of the ground-truth vector is $\frac{1}{N}\mathbf{w}^{*\mathsf{T}}\mathbf{w}^* = 1$. Moreover, $L$ represents length of the original weight vector $\mathbf{w}$ and $L^2 = \frac{1}{N}\mathbf{w}^{\mathsf{T}}\mathbf{w}$.

**Learning Dynamics.** The update equations (5) can be transformed into a set of differential equations (ODEs) by using the above order parameters. This is achieved by treating the update step $j$ as a continuous time variable $t = \frac{j}{N}$. They can be turned into differential equations because the contiguous step $\Delta t = \frac{1}{N}$ approaches zero in the thermodynamic limit when $N \to \infty$. We obtain a dynamical system of three order parameters

$$\frac{d\gamma}{dt} = \eta\frac{I_1}{\gamma} - \eta\zeta\gamma, \quad \frac{dR}{dt} = \eta\frac{\gamma}{L^2}I_3 - \eta\frac{R}{L^2}I_1 - \eta^2\frac{\gamma^2 R}{2L^4}I_2, \quad \text{and} \quad \frac{dL}{dt} = \eta^2\frac{\gamma^2}{2L^3}I_2, \qquad (6)$$

where $I_1 = \mathbb{E}_\mathbf{x}[\delta\tilde{\mathbf{w}}^{\mathsf{T}}\mathbf{x}]$, $I_2 = \mathbb{E}_\mathbf{x}[\delta^2\mathbf{x}^{\mathsf{T}}\mathbf{x}]$, and $I_3 = \mathbb{E}_\mathbf{x}[\delta\mathbf{w}^{*\mathsf{T}}\mathbf{x}]$ are defined to simplify notations. The derivations of Eqn.(6) can be found in Appendix C.5.

## 3.1 FIXED POINT OF THE DYNAMICAL SYSTEM

To find the fixed points of (6), we set $d\gamma/dt = dR/dt = dL/dt = 0$. The fixed points of BN, WN, and vanilla SGD (without BN and WN) are given in Table 1. In the thermodynamic limit, the optima denoted as $(\gamma_0, R_0, L_0)$ would be $(\gamma_0, R_0, L_0) = (1, 1, 1)$. Our main interest is the overlapping ratio $R_0$ between the student and the teacher, because it optimizes the direction of the weight vector regardless of its length. We see that $R_0$ for all three approaches attain optimum '1'. Intuitively, in BN and WN, this optimal solution

|  | $(\gamma_0, R_0, L_0)$ | $\eta_{\max}$ $(R)$ | $\eta_{\text{eff}}$ $(R)$ |
|---|---|---|---|
| BN | $(\gamma_0, 1, L_0)$ | $(\frac{\partial(\gamma_0 I_3 - I_1)}{\gamma_0 \partial R} - \zeta\gamma_0)/\frac{\partial I_2}{2\partial R}$ | $\frac{\eta\gamma_0}{L_0^2}$ |
| WN | $(1, 1, L_0)$ | $\frac{\partial(I_3 - I_1)}{\partial R}/\frac{\partial I_2}{2\partial R}$ | $\frac{\eta}{L_0^2}$ |
| SGD | $(1, 1, 1)$ | $\frac{\partial(I_3 - I_1)}{\partial R}/\frac{\partial I_2}{2\partial R}$ | $\eta$ |

Table 1: Comparisons of fixed points, $\eta_{\max}$ for $R$, and $\eta_{\text{eff}}$ for $R$. A fixed point is denoted as $(\gamma_0, R_0, L_0)$.

does not depend on the value of $L_0$ because their weight vectors are normalized. In other words, WN and BN are easier to optimize than vanilla SGD, unlike SGD where both $R_0$ and $L_0$ have to be optimized to '1'. Furthermore, $\gamma_0$ in BN depends on the activation function. For ReLU, we have $\gamma_0^{bn} = \frac{1}{2\zeta+1}$ (see Proposition 1 in Appendix C.5), meaning that norm of the normalized weight vector relies on the decay factor $\zeta$. In WN, we have $\gamma_0^{wn} = 1$ as WN has no regularization on $\gamma$.

## 3.2 MAXIMUM AND EFFECTIVE LEARNING RATES

With the above fixed points, we derive the maximum and the effective LR. Specifically, we analyze eigenvalues and eigenvectors of the Jacobian matrix corresponding to Eqn.(6). We are interested in the LR to approach $R_0$. We find that this optimum value only depends on its corresponding eigenvalue denoted as $\lambda_R$. We have $\lambda_R = \frac{\partial I_2}{\partial R}\frac{\eta\gamma_0}{2L_0^2}(\eta_{\max} - \eta_{\text{eff}})$, where $\eta_{\max}$ and $\eta_{\text{eff}}$ represent the maximum and effective LR (proposition 2 in Appendix C.5), which are given in Table 1. We demonstrate that $\lambda_R < 0$ if and only if $\eta_{\max} > \eta_{\text{eff}}$, such that the fixed point $R_0$ is stable for all approaches (proposition 3 in Appendix C.6). Moreover, it is also able to show that $\eta_{\max}$ of BN ($\eta_{\max}^{bn}$) is larger than WN and SGD, enabling $R$ to converge with a larger learning rate. For ReLU as an example, we find that $\eta_{\max}^{bn} \geq \eta_{\max}^{\{wn,sgd\}} + 2\zeta$ (proposition 4 in Appendix C.7). The larger maximum LRs enables the network to be trained more stably and has the potential to be combined with other stabilization techniques (Fagan & Iyengar, 2018) during optimization. The effective LRs shown in Table 1 are consistent with previous work (van Laarhoven, 2017).

---

[1]$g'(x)$ denotes the first derivative of $g(x)$.

# 4 GENERALIZATION ANALYSIS

Here we investigate generalization of BN by using a teacher-student model that minimizes a loss function $\frac{1}{P}\sum_{j=1}^{P}((y^*)^j - y^j)^2$, where $y^*$ represents the teacher's output and $y$ is the student's output. We compare BN with WN+gamma decay and vanilla SGD. All of them share the same teacher network whose output is a noise-corrupted linear function $y^* = \mathbf{w}^{*\mathsf{T}}\mathbf{x} + s$, where $\mathbf{x}$ is drawn from $\mathcal{N}(0, \frac{1}{N})$ and $s$ is an unobserved Gaussian noise. We are interested to see how the above methods resist this noise by using student networks with both identity (linear) and ReLU activation functions.

For **vanilla SGD**, the student is computed by $y = g(\mathbf{w}^{\mathsf{T}}\mathbf{x})$ with $g(\cdot)$ being either identity or ReLU, and $\mathbf{w}$ being the weight vector to optimize, where $\mathbf{w}$ has the same dimension as $\mathbf{w}^*$. The loss function of vanilla SGD is $\ell^{sgd} = \frac{1}{P}\sum_{j=1}^{P}\left(y^* - g(\mathbf{w}^{\mathsf{T}}\mathbf{x}^j)\right)^2$. For **BN**, the student is defined as $y = \gamma\frac{\mathbf{w}^{\mathsf{T}}\mathbf{x} - \mu_{\mathcal{B}}}{\sigma_{\mathcal{B}}} + \beta$. As our main interest is the weight vector, we freeze the bias by setting $\beta = 0$. Therefore, the batch average term $\mu_{\mathcal{B}}$ is also unnecessary to avoid additional parameters, and the loss function is written as $\ell^{bn} = \frac{1}{P}\sum_{j=1}^{P}\left((y^*)^j - \gamma\mathbf{w}^{\mathsf{T}}\mathbf{x}^j/\sigma_{\mathcal{B}}\right)^2$. For **WN+gamma decay**, the student is computed similar to Eqn.(4) by using $y = \sqrt{N}\gamma\frac{\mathbf{w}^{\mathsf{T}}\mathbf{x}}{\|\mathbf{w}\|_2}$. Then the loss function is defined by $\ell^{wn} = \frac{1}{P}\sum_{j=1}^{P}\left((y^*)^j - \sqrt{N}\gamma\frac{\mathbf{w}^{\mathsf{T}}\mathbf{x}^j}{\|\mathbf{w}\|_2}\right)^2 + \zeta\|\gamma\|_2^2$. With the above definitions, the three approaches are studied under the same teacher-student framework, where their generalization errors can be strictly compared with the other factors ruled out.

## 4.1 GENERALIZATION ERRORS

We provide closed-form solutions of the generalization errors (see Appendix D.1) for vanilla SGD with both linear and ReLU student networks. The theoretical solution of WN+gamma decay can also be solved for the linear student, but still remains difficult for ReLU student whose numerical verification is provided instead. Both vanilla SGD and WN+gamma decay are compared with numerical solutions of BN.

**vanilla SGD.** In an identity (linear) student, the solution of generalization error depends on the rank of correlation matrix $\mathbf{\Sigma} = \mathbf{x}^{\mathsf{T}}\mathbf{x}$. Here we define an effective load $\alpha = P/N$ that is the ratio between number of samples $P$ and number of input neurons $N$ (number of learnable parameters).

The generalization error of the identity student is denoted as $\epsilon_{\mathrm{id}}^{sgd}$, which can be acquired by using the distribution of eigenvalues of $\mathbf{\Sigma}$ following (Advani & Saxe, 2017). If $\alpha < 1$, $\epsilon_{\mathrm{id}}^{sgd} = 1 - \alpha + \alpha S/(1-\alpha)$. Otherwise, $\epsilon_{\mathrm{id}}^{sgd} = S/(\alpha-1)$ where $S$ is the variance of the injected noise to the teacher network. The values of $\epsilon_{\mathrm{id}}^{sgd}$ with respect to $\alpha$ are plotted in blue curve of Fig.1(a). It first decreases but then increases as $\alpha$ increases from 0 to 1. $\epsilon_{\mathrm{id}}^{sgd}$ diverges at $\alpha = 1$. And it would decrease again when $\alpha > 1$.

In a ReLU student, the nonlinear activation yields difficulty to derive the theoretical solution. Here we utilize the statistical mechanics and calculate that $\epsilon_{\mathrm{relu}}^{sgd} = 1 - \alpha/4 + \frac{\alpha S}{2(2-\alpha)}$ and $\alpha < 2$ (see AppendixD.2). When comparing to the lower bound (trained without noisy supervision) shown as the red curve in Fig.1(b), we see that $\epsilon_{\mathrm{relu}}^{sgd}$ (blue curve) diverges at $\alpha = 2$. This is because the student overfits the noise in the teacher's output. The curve of numerical solution is also plotted in dashed line in Fig.1(b) and it captures the diverging trend well. It should be noted that obtaining the theoretical curve empirically requires

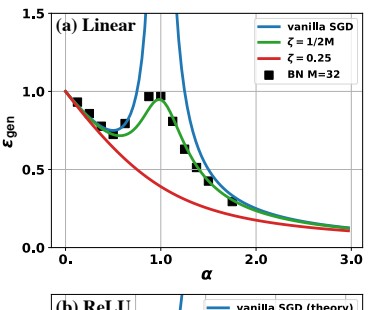

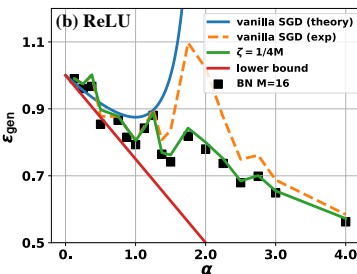

Figure 1: **(a)** shows generalization error *v.s.* effective load $\alpha$ using a linear student (identity units). 'WN+gamma decay' has two curves $\zeta = \frac{1}{2M}$ and $\zeta = 0.25$. BN is trained with $M = 32$. **(b)** shows generalization error *v.s.* effective load $\alpha$ using a ReLU student. 'WN+gamma decay' has $\zeta = \frac{1}{4M}$ and is compared to BN with batch size $M = 32$. The theoretical curve for vanilla SGD is also shown in blue. The red line is the generalization error of vanilla SGD with no noise in the teacher and thus serves as a lower bound.

an infinitely long time of training and an infinitely small learning rate. This unreachable limit explains the discrepancies between the theoretical and the numerical solution.

**WN+gamma decay.** In a linear student, the gamma decay term turns the correlation matrix to $\Sigma = (\mathbf{x}^\mathsf{T}\mathbf{x} + \zeta\mathbf{I})$, which is positive definite. Following statistical mechanics (Krogh & Hertz, 1992), the generalization error is $\epsilon_{\mathrm{id}}^{wn} = \delta^2\frac{\partial(\zeta G)}{\partial\zeta} - \zeta^2\frac{\partial G}{\partial\zeta}$ where $G = 1 - \alpha - \zeta + (\zeta + (1+\sqrt{\alpha})^2)^{\frac{1}{2}}(\zeta + (1-\sqrt{\alpha})^2)^{\frac{1}{2}}/2\zeta$. We see that $\epsilon_{\mathrm{id}}^{wn}$ can be computed quantitatively given the values of $\zeta$ and $\alpha$. Let the variance of noise injected to the teacher be $0.25$. Fig.1(a) shows that no other curves could outperform the red curve when $\zeta = 0.25$, a value equal to the noise magnitude. The $\zeta$ smaller than $0.25$ (green curve $\zeta = \frac{1}{2M}$ and $M = 32$) would exhibit overtraining around $\alpha = 1$, but they still perform significantly better than vanilla SGD.

For the ReLU student in Fig,1(b), a direct solution of the generalization error $\epsilon_{\mathrm{relu}}^{wn}$ remains an open problem. Therefore, the numerical results of 'WN+gamma decay' (green curve) are run at each $\alpha$ value. It effectively reduces over-fitting compared to vanilla SGD.

**Numerical Solutions of BN.** In the linear student, we employ SGD with $M = 32$ to find solutions of $\mathbf{w}$ for BN. The number of input neurons is 4096 and the number of training samples can be varied to change $\alpha$. The results are marked as black squares in Fig.1(a). After applying the analyses for linear student (Appendix C.3), BN is equivalent to 'WN+gamma decay' when $\zeta = \frac{1}{2M}$ (green curve). It is seen that BN gets in line with the curve '$\zeta = 1/2M$' ($M = 32$) and thus quantitatively validates our derivations.

In the ReLU student, the setting is mostly the same as the linear case, except that we employ a smaller batch size $M = 16$. The results are shown as black squares in Fig.1(b). For ReLU units, the equivalent $\zeta$ of gamma decay is $\zeta = \frac{1}{4M}$. If one compares the generalization error of BN with 'WN+gamma decay' (green curve), a clear correspondence is found, which also validates the derivations for the ReLU activation function.

## 5 EXPERIMENTS IN CNNs

This section shows that BN in CNNs follows similar traits of regularization as the above analyses.

To compare different methods, the CNN architectures are fixed while only the normalization layers are changed. We adopt CIFAR10 (Krizhevsky, 2009) that contains 60k images of 10 categories (50k images for training and 10k images for test). All models are trained by using SGD with momentum, while the initial learning rates are scaled proportionally (Goyal et al., 2017) when different batch sizes are presented. More empirical setting can be found in Appendix B.

**Evaluation of PN+Gamma Decay**. This work shows that BN can be decomposed into PN and gamma decay. We empirically compare 'PN+gamma decay' with BN by using ResNet18 (He et al., 2016). For 'PN+gamma decay', the population statistics of PN and the decay factor of gamma decay are estimated by using sufficient amount of training samples. For BN, BN trained with a normal batch size $M = 128$ is treated as baseline as shown in Fig.2(a&b). We see that when batch size increases, BN would imped both loss and accuracy. For example, when increasing $M$ to 1024, performance decreases because the regularization from the batch statistics reduces in large batch, resulting in overtraining (see the gap between train and validation loss in (a) when $M = 1024$).

In comparison, we train PN by using 10k training samples to estimate the population statistics. Note that this further reduces regularization. We see that the release of regularization can be complemented by gamma decay, making PN outperformed BN. This empirical result verifies our derivation of regularization for BN. Similar trend can be observed by experiment in a down-sampled version of ImageNet (see Appendix B.1). We would like to point out that 'PN+gamma decay' is of interest in theoretical analyses, but it is computation-demanding when applied in practice because evaluating $\mu_\mathcal{P}$, $\sigma_\mathcal{P}$ and $\zeta(h)$ may require sufficiently large number of samples.

**Comparisons of Regularization.** We study the regulation strengths of vanilla SGD, BN, WN, WN+mean-only BN, and WN+variance-only BN. At first, the strength of regularization terms from both $\mu_\mathcal{B}$ and $\sigma_\mathcal{B}$ are compared by using a simpler network with 4 convolutional and 2 fully connected layers as used in (Salimans & Kingma, 2016). Fig.2(c&d) compares their training and validation losses. We see that the generalization error of BN is much lower than WN and vanilla SGD. The reason has been disclosed in this work: stochastic behaviors of $\mu_\mathcal{B}$ and $\sigma_\mathcal{B}$ in BN improves generalization.

To investigate $\mu_\mathcal{B}$ and $\sigma_\mathcal{B}$ individually, we decompose their contributions by running a WN with mean-only BN as well as a WN with variance-only BN, to simulate their respective regularization.

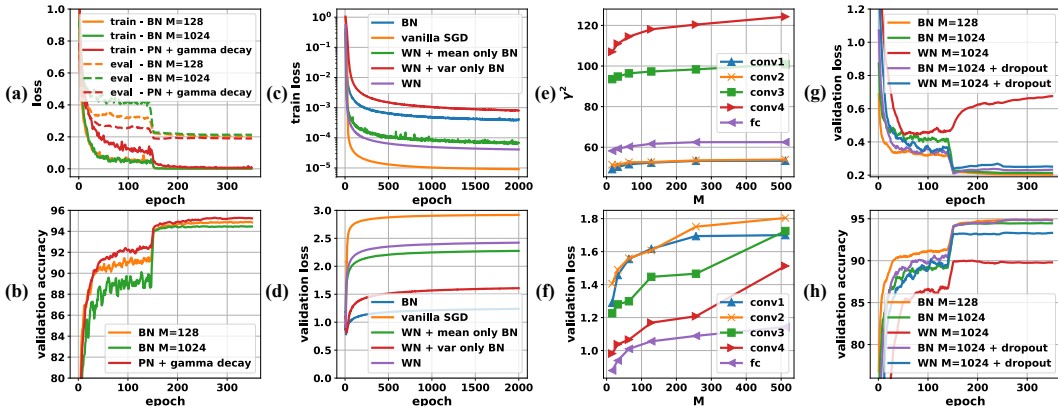

Figure 2: (a) & (b) compare the loss (both training and evaluation) and validation accuracy between BN and PN on CIFAR10 using a ResNet18 network; (c) & (d) compare the training and validation loss curve with WN + mean-only BN and WN + variance-only BN; (e) & (f) validate the regularization effect of BN on both $\gamma^2$ and the validation loss with different batch sizes; (g) & (h) show the loss and top-1 validation accuracy of ResNet18 with additional regularization (dropout) on large-batch training of BN and WN.

As shown in Fig.2(c&d), improvements from the mean-only and the variance-only BN over WN verify our conclusion that noises from $\mu_{\mathcal{B}}$ and $\sigma_{\mathcal{B}}$ have different regularization strengths. Both of them are essential to produce good result.

**Regularization and parameter norm.** We further demonstrate impact of BN to the norm of parameters. We compare BN with vanilla SGD. A network is first trained by BN in order to converge to a local minima where the parameters do not change much. At this local minima, the weight vector is frozen and denoted as $\mathbf{w}^{bn}$. Then this network is finetuned by using vanilla SGD with a small learning rate $10^{-3}$ and its kernel parameters are initialized by $\mathbf{w}^{sgd} = \gamma \frac{\mathbf{w}^{bn}}{\sigma}$, where $\sigma$ is the moving average of $\sigma_{\mathcal{B}}$.

Fig.4 in Appendix B.2 visualizes the results. As $\mu_{\mathcal{B}}$ and $\sigma_{\mathcal{B}}$ are removed in vanilla SGD, it is found that the training loss decreases while the validation loss increases, implying that reduction in regularization makes the network converged to a sharper local minimum that generalizes less well. The magnitudes of kernel parameters $\mathbf{w}^{sgd}$ at different layers are also observed to increase after freezing BN, due to the release of regularization on these parameters.

**Batch size.** To study BN with different batch sizes, we train different networks but only add BN at one layer at a time. The regularization on the $\gamma$ parameter is compared in Fig.2(e) when BN is located at different layers. The values of $\gamma^2$ increase along with the batch size $M$ due to the weaker regularization for the larger batches. The increase of $\gamma^2$ also makes all validation losses increased as shown in Fig.2(f).

**BN and WN trained with dropout**. As PN and gamma decay requires estimating the population statistics that increases computations, we utilize dropout as an alternative to improve regularization of BN. We add a dropout after each BN layer. Fig.2(g&h) plot the classification results using ResNet18. The generalization of BN deteriorates significantly when $M$ increases from 128 to 1024. This is observed by the much higher validation loss (Fig.2(g)) and lower validation accuracy (Fig.2(h)) when $M = 1024$. If a dropout layer with ratio 0.1 is added *after* each residual block layer for $M = 1024$ in ResNet18, the validation loss is suppressed and accuracy increased by a great margin. This superficially contradicts with the original claim that BN reduces the need for dropout (Ioffe & Szegedy, 2015). As discussed in Appendix B.3, we find that there are two differences between our study and previous work (Ioffe & Szegedy, 2015).

Fig.2(g&h) also show that WN can also be regularized by dropout. We apply dropout after each WN layer with ratio 0.2 and the dropout is applied at the same layers as that for BN. We found that the improvement on both validation accuracy and loss is surprising. The accuracy increases from 0.90 to 0.93, even close to the results of BN. Nevertheless, additional regularization on WN still cannot make WN on par with the performance BN. In deep neural networks the distribution after each layer would be far from a Gaussian distribution, in which case WN is not a good substitute for PN.

## 6    CONCLUSIONS

This work investigated an explicit regularization form of BN, which was decomposed into PN and gamma decay where the regularization strengths from $\mu_\mathcal{B}$ and $\sigma_\mathcal{B}$ were explored. Moreover, optimization and generalization of BN with regularization were derived and compared with vanilla SGD, WN, and WN+gamma decay, showing that BN enables training to converge with large maximum and effective learning rate, as well as leads to better generalization. Our analytical results explain many existing empirical phenomena. Experiments in CNNs showed that BN in deep networks share the same traits of regularization. In future work, we are interested in analyzing optimization and generalization of BN in deep networks, which is still an open problem. Moreover, investigating the other normalizers such as instance normalization (IN) (Ulyanov et al., 2016) and layer normalization (LN) (Ba et al., 2016) is also important. Understanding the characteristics of these normalizers should be the first step to analyze some recent best practices such as whitening (Luo, 2017b;a), switchable normalization (Luo et al., 2019; 2018; Shao et al., 2019), and switchable whitening (Pan et al., 2019).

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

## APPENDICES

## A  NOTATIONS

Table 2: Several notations are summarized for reference.

| | |
|---|---|
| $\mu_{\mathcal{B}}, \sigma_{\mathcal{B}}^2$ | batch mean, batch variance |
| $\mu_{\mathcal{P}}, \sigma_{\mathcal{P}}^2$ | population mean, population variance |
| $\mathbf{x}, y$ | input of a network, output of a network |
| $y^*$ | ground truth of an output |
| $h, \hat{h}$ | hidden value before and after BN |
| $\check{h}$ | hidden value after population normalization |
| $\gamma, \beta$ | scale parameter, shift parameter |
| $g(\cdot)$ | activation function |
| $\mathbf{w}, \mathbf{w}^*$ | weight vector, ground truth weight vector |
| $\tilde{\mathbf{w}}$ | normalized weight vector |
| $M, N, P$ | batch size, number of neurons, sample size |
| $\alpha$ | an effective load value $\alpha = P/N$ |
| $\zeta$ | regularization strength (coefficient) |
| $\rho$ | Kurtosis of a distribution |
| $\delta$ | gradient of the activation function |
| $\eta_{\text{eff}}, \eta_{\text{max}}$ | effective, maximum learning rate |
| $R$ | overlapping ratio (angle) between $\tilde{\mathbf{w}}$ and $\mathbf{w}^*$ |
| $L$ | norm (length) of $\mathbf{w}$ |
| $\lambda_{\text{max}}, \lambda_{\text{min}}$ | maximum, minimum eigenvalue |
| $\epsilon_{\text{gen}}$ | generalization error |

## B  MORE EMPIRICAL SETTINGS AND RESULTS

All experiments in Sec.5 are conducted in CIFAR10 by using ResNet18 and a CNN architecture similar to (Salimans & Kingma, 2016) that is summarized as 'conv(3,32)-conv(3,32)-conv(3,64)-conv(3,64)-pool(2,2)-fc(512)-fc(10)', where 'conv(3,32)' represents a convolution with kernel size 3 and 32 channels, 'pool(2,2)' is max-pooling with kernel size 2 and stride 2, and 'fc' indicates a full connection. We follow a configuration for training by using SGD with a momentum value of 0.9 and continuously decaying the learning rate by a factor of $10^{-4}$ each step. For different batch sizes, the initial learning rate is scaled proportionally with the batch size to maintain a similar learning dynamics (Goyal et al., 2017).

### B.1  RESULTS IN DOWNSAMPLED IMAGENET

Besides CIFAR10, we also evaluate 'PN+gamma decay' by employing a downsampled version of ImageNet (Loshchilov & Hutter, 2016), which contains identical 1.2 million data and 1k categories as the original ImageNet, but each image is scaled to $32 \times 32$. We train ResNet18 in downsampled ImageNet by following the training protocol used in (He et al., 2016). In particular, ResNet18 is trained by using SGD with momentum of 0.9 and the initial learning rate is 0.1, which is then decayed by a factor of 10 after 30, 60, and 90 training epochs.

In downsampled ImageNet, we observe similar trends as those presented in CIFAR10. For example, we see that BN would imped both loss and accuracy when batch size increases. When increasing $M$ to 1024 as shown in Fig.3, both the loss and validation accuracy decrease because the regularization from the random batch statistics reduces in large batch size, resulting in overtraining. This can be seen by the gap between the training and the validation loss. Nevertheless, we see that the reduction of regularization can be complemented when PN is trained with adaptive gamma decay, which makes PN performed comparably to BN in downsampled ImageNet.

### B.2  IMPACT OF BN TO THE NORM OF PARAMETERS

We demonstrate the impact of BN to the norm of parameters. We compare BN with vanilla SGD, where a network is first trained by BN in order to converge to a local minima when the parameters do not change much. At this local minima, the weight vector is frozen and denoted as $\mathbf{w}^{bn}$. Then this

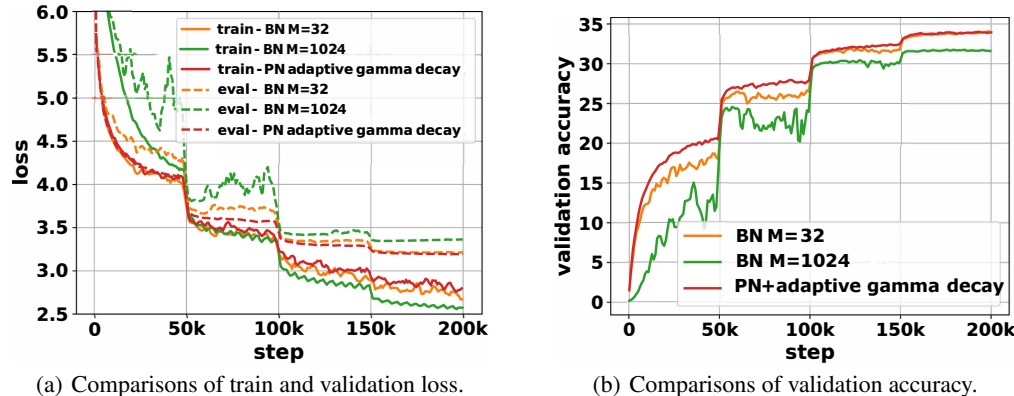

(a) Comparisons of train and validation loss.

(b) Comparisons of validation accuracy.

Figure 3: **Results of downsampled ImageNet.** (a) plots training and evaluation loss. (b) shows validation accuracy. The models are trained on 8 GPUs.

network is finetuned by using vanilla SGD with a small learning rate $10^{-3}$ with the kernel parameters initialized by $\mathbf{w}^{sgd} = \gamma \frac{\mathbf{w}^{bn}}{\sigma}$, where $\sigma$ is the moving average of $\sigma_{\mathcal{B}}$.

Fig.4 below visualizes the results. As $\mu_{\mathcal{B}}$ and $\sigma_{\mathcal{B}}$ are removed in the vanilla SGD, it is found from the last two figures that the training loss decreases while the validation loss increases, meaning that the reduction in regularization makes the network converged to a sharper local minimum that generalizes less well. The magnitudes of kernel parameters $\mathbf{w}^{sgd}$ at different layers are also displayed in the first four figures. All of them increase after freezing BN, due to the release of regularization on these parameters.

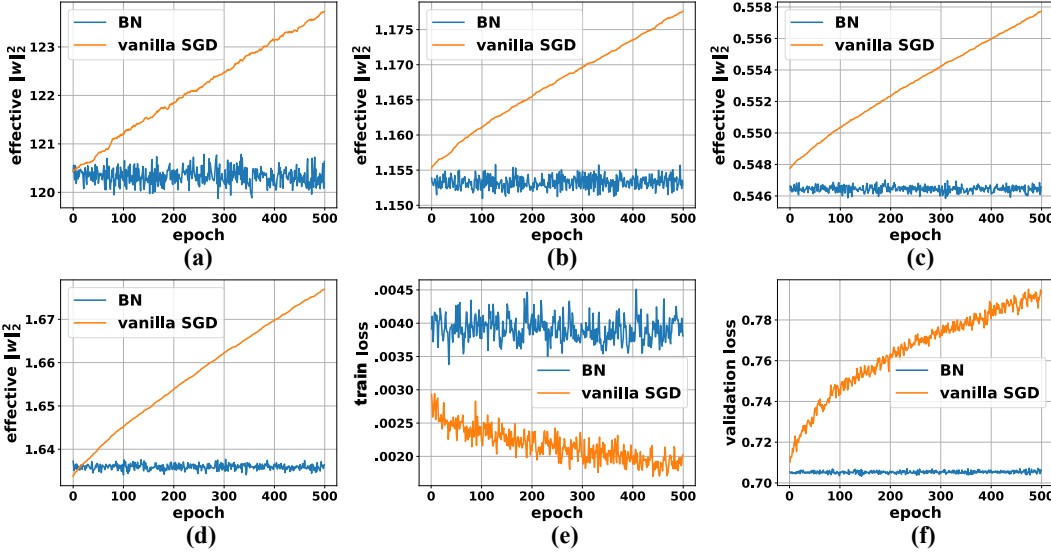

Figure 4: **Study of parameter norm.** Vanilla SGD is finetuned from a network pretrained by BN on CIFAR10. The first four figures show the magnitude of the kernel parameters in different layers in finetuning, compared to the effective norm of BN defined as $\gamma \frac{\|\mathbf{w}\|}{\sigma_{\mathcal{B}}}$. The last two figures compare the training and validation losses in finetuning.

### B.3    BN AND WN WITH DROPOUT

**BN+dropout**.    Despite the better generalization of BN with smaller batch sizes, large-batch training is more efficient in real cases. Therefore, improving generalization of BN with large batch is more desiring. However, gamma decay requires estimating the population statistics that

increases computations. We also found that treating the decay factor as a constant hardly improves generalization for large batch. Therefore, we utilize dropout as an alternative to compensate for the insufficient regularization. Dropout has also been analytically viewed as a regularizer (Wager et al., 2013). We add a dropout after each BN layer to impose regularization.

Fig.2(g&h) in the main paper plot the classification results using ResNet18. The generalization of BN deteriorates significantly when $M$ increases from 128 to 1024. This is observed by the much higher validation loss (Fig.2(g)) and lower validation accuracy (Fig.2(h)) when $M = 1024$. If a dropout layer with ratio 0.1 is added *after* each residual block layer for $M = 1024$ in ResNet18, the validation loss is suppressed and accuracy increased by a great margin. This superficially contradicts with the original claim that BN reduces the need for dropout (Ioffe & Szegedy, 2015). We find that there are two differences between our study and (Ioffe & Szegedy, 2015).

First, in pervious study the batch size was fixed at a quite small value (*e.g.* 32), at which the regularization was already quite strong. Therefore, an additional dropout could not further cause better regularization, but on the contrary increases the instability in training and yields a lower accuracy. However, our study explores relatively large batch that degrades the regularization of BN, and thus dropout with a small ratio can complement. Second, usual trials put dropout before BN and cause BN to have different variances during training and test. In contrast, dropout follows BN in this study and the distance between two dropout layers is large (a residual block separation), thus the problem can be alleviated. The improvement by applying dropout after BN has also been observed by a recent work (Li et al., 2018).

**WN+dropout**. Since BN can be treated as WN trained with regularization as shown in this study, combining WN with regularization should be able to match the performance of BN. As WN outperforms BN in running speed (without calculating statistics) and it suits better in RNNs than BN, an improvement of its generalization is also of great importance. Fig.2(g&h) also show that WN can also be regularized by dropout. We apply dropout after each WN layer with ratio 0.2 and the dropout is applied at the same layers as that for BN. We found that the improvement on both validation accuracy and loss is surprising. The accuracy increases from 0.90 to 0.93, even close to the results of BN. Nevertheless, additional regularization on WN still cannot make WN on par with the performance BN. In deep neural networks the distribution after each layer would be far from a Gaussian distribution, in which case WN is not a good substitute for PN. A potential substibute of BN would require us for designing better estimations of the distribution to improve the training speed and performance of deep networks.

## C  PROOF OF RESULTS

### C.1  PROOF OF EQN.(3)

**Theorem 1** (Regularization of $\mu_\mathcal{B}, \sigma_\mathcal{B}$). *Let a single-layer perceptron with BN and ReLU activation function be defined by $y = \max(0, \hat{h})$, $\hat{h} = \gamma \frac{h - \mu_\mathcal{B}}{\sigma_\mathcal{B}} + \beta$ and $h = \mathbf{w}^\mathsf{T}\mathbf{x}$, where $\mathbf{x}$ and $y$ are the network input and output respectively, $h$ and $\hat{h}$ are the hidden values before and after batch normalization, and $\mathbf{w}$ is the weight vector. Let $\ell(\hat{h})$ be the loss function. Then*

$$\frac{1}{P}\sum_{j=1}^{P}\mathbb{E}_{\mu_\mathcal{B}, \sigma_\mathcal{B}}\ell(\hat{h}^j) \simeq \frac{1}{P}\sum_{j=1}^{P}\ell(\bar{h}^j) + \zeta(h)\gamma^2 \ \text{ and } \ \zeta(h) = \frac{\rho + 2}{8M}\mathcal{I}(\gamma) + \frac{1}{2M}\frac{1}{P}\sum_{j=1}^{P}\sigma(\bar{h}^j),$$

*where $\bar{h}^j = \gamma\frac{\mathbf{w}^\mathsf{T}\mathbf{x}^j - \mu_\mathcal{P}}{\sigma_\mathcal{P}} + \beta$ represents population normalization (PN), $\zeta(h)\gamma^2$ represents gamma decay and $\zeta(h)$ is a data-dependent decay factor. $\rho$ is the kurtosis of the distribution of $h$, $\mathcal{I}(\gamma)$ is an estimation of the Fisher information of $\gamma$ and $\mathcal{I}(\gamma) = \frac{1}{P}\sum_{j=1}^{P}(\frac{\partial \ell(\hat{h}^j)}{\partial \gamma})^2$, and $\sigma(\cdot)$ is a sigmoid function.*

*Proof.* We have $\hat{h}^j = \gamma\frac{\mathbf{w}^T\mathbf{x}^j - \mu_\mathcal{B}}{\sigma_\mathcal{B}} + \beta$ and $\bar{h}^j = \gamma\frac{\mathbf{w}^T\mathbf{x}^j - \mu_\mathcal{P}}{\sigma_\mathcal{P}} + \beta$. We prove theorem 1 by performing a Taylor expansion on a function $A(\hat{h}^j)$ at $\bar{h}^j$, where $A(\hat{h}^j)$ is a function of $\hat{h}^j$ defined according to a particular activation function. The negative log likelihood function of the above single-layer perceptron can be generally defined as $-\log p(y^j|\hat{h}^j) = A(\hat{h}^j) - y^j\hat{h}^j$, which is similar to the loss

function of the generalized linear models with different activation functions. Therefore, we have

$$
\frac{1}{P}\sum_{j=1}^{P}\mathbb{E}_{\mu_{\mathcal{B}},\sigma_{\mathcal{B}}}[l(\hat{h}^j)] = \frac{1}{P}\sum_{j=1}^{P}\mathbb{E}_{\mu_{\mathcal{B}},\sigma_{\mathcal{B}}}\left[A(\hat{h}^j) - y^j\hat{h}^j\right]
$$

$$
= \frac{1}{P}\sum_{j=1}^{P}(A(\bar{h}^j) - y^j\bar{h}^j) + \frac{1}{P}\sum_{j=1}^{P}\mathbb{E}_{\mu_{\mathcal{B}},\sigma_{\mathcal{B}}}\left[-y^j(\hat{h}^j - \bar{h}^j) + A(\hat{h}^j) - A(\bar{h}^j)\right]
$$

$$
= \frac{1}{P}\sum_{j=1}^{P}l(\bar{h}^j) + \frac{1}{P}\sum_{j=1}^{P}\mathbb{E}_{\mu_{\mathcal{B}},\sigma_{\mathcal{B}}}\left[(A'(\bar{h}^j) - y^j)(\hat{h}^j - \bar{h}^j)\right]
$$

$$
+ \frac{1}{P}\sum_{j=1}^{P}\mathbb{E}_{\mu_{\mathcal{B}},\sigma_{\mathcal{B}}}\left[\frac{A''(\bar{h}^j)}{2}(\hat{h}^j - \bar{h}^j)^2\right]
$$

$$
= \frac{1}{P}\sum_{j=1}^{P}l(\bar{h}^j) + R^f + R^q,
$$

where $A'(\cdot)$ and $A''(\cdot)$ denote the first and second derivatives of function $A(\cdot)$. The first and second order terms in the expansion are represented by $R^f$ and $R^q$ respectively. To derive the analytical forms of $R^f$ and $R^q$, we take a second-order Taylor expansion of of $\frac{1}{\sigma_{\mathcal{B}}}$ and $\frac{1}{\sigma_{\mathcal{B}}^2}$ around $\sigma_P$, it suffices to have

$$
\frac{1}{\sigma_{\mathcal{B}}} \approx \frac{1}{\sigma_{\mathcal{P}}} + (-\frac{1}{\sigma_{\mathcal{P}}^2})(\sigma_{\mathcal{B}} - \sigma_{\mathcal{P}}) + \frac{1}{\sigma_{\mathcal{P}}^3}(\sigma_{\mathcal{B}} - \sigma_{\mathcal{P}})^2
$$

and

$$
\frac{1}{\sigma_{\mathcal{B}}^2} \approx \frac{1}{\sigma_{\mathcal{P}}^2} + (-\frac{2}{\sigma_{\mathcal{P}}^3})(\sigma_{\mathcal{B}} - \sigma_{\mathcal{P}}) + \frac{3}{\sigma_{\mathcal{P}}^4}(\sigma_{\mathcal{B}} - \sigma_{\mathcal{P}})^2.
$$

By applying the distributions of $\mu_{\mathcal{B}}$ and $\sigma_{\mathcal{B}}$ introduced in section 2, we have $\mu_{\mathcal{B}} \sim \mathcal{N}(\mu_{\mathcal{P}}, \frac{\sigma_{\mathcal{P}}^2}{M})$ and $\sigma_{\mathcal{B}} \sim \mathcal{N}(\sigma_P, \frac{\rho+2}{4M})$. Hence, $R^f$ can be derived as in the paper, $R^f$ can be derived as

$$
R^f = \frac{1}{P}\sum_{j=1}^{P}\mathbb{E}_{\mu_{\mathcal{B}},\sigma_{\mathcal{B}}}\left[(A'(\bar{h}^j) - y^j)(\hat{h}^j\bar{h}^j)\right]
$$

$$
= \frac{1}{P}\sum_{j=1}^{P}\mathbb{E}_{\mu_{\mathcal{B}},\sigma_{\mathcal{B}}}\left[(A'(\bar{h}^j) - y^j)\left(\gamma\frac{\mathbf{w}^T\mathbf{x}^j - \mu_{\mathcal{B}}}{\sigma_{\mathcal{B}}} - \gamma\frac{\mathbf{w}^T\mathbf{x}^j - \mu_{\mathcal{P}}}{\sigma_{\mathcal{P}}}\right)\right]
$$

$$
= \frac{1}{P}\sum_{j=1}^{P}\mathbb{E}_{\mu_{\mathcal{B}},\sigma_{\mathcal{B}}}\left[(A'(\bar{h}^jy^j)\left(\gamma\mathbf{w}^T\mathbf{x}^j\left(\frac{1}{\sigma_{\mathcal{B}}} - \frac{1}{\sigma_{\mathcal{P}}}\right) + \gamma\left(-\frac{\mu_{\mathcal{B}}}{\sigma_{\mathcal{B}}} + \frac{\mu_{\mathcal{P}}}{\sigma_{\mathcal{P}}}\right)\right)\right]
$$

$$
= \frac{1}{P}\sum_{j=1}^{P}\gamma(A'(\bar{h}^j) - y^j)(\mathbf{w}^T\mathbf{x}^j - \mu_{\mathcal{P}})\mathbb{E}_{\sigma_{\mathcal{B}}}\left[\frac{1}{\sigma_{\mathcal{B}}} - \frac{1}{\sigma_{\mathcal{P}}}\right]
$$

$$
= \frac{1}{P}\sum_{j=1}^{P}\frac{\rho+2}{4M}\gamma(A'(\bar{h}^j) - y^j)\frac{\mathbf{w}^T\mathbf{x}^j - \mu_{\mathcal{P}}}{\sigma_{\mathcal{P}}}.
$$

This $R^f$ term can be understood as below. Let $h = \frac{\mathbf{w}^T\mathbf{x} - \mu_{\mathcal{P}}}{\sigma_{\mathcal{P}}}$ and the distribution of the population data be $p_{xy}$. We establish the following relationship

$$
\mathbb{E}_{(x,y)\sim p_{xy}}\mathbb{E}_{\mu_{\mathcal{B}},\sigma_{\mathcal{B}}}\left[(A'(\bar{h}) - y)h\right] = \mathbb{E}_{\mu_{\mathcal{B}},\sigma_{\mathcal{B}}}\mathbb{E}_{x\sim p_x}\mathbb{E}_{y|x\sim p_{y|x}}\left[(A'(\bar{h}) - y)h\right]
$$

$$
= \mathbb{E}_{\mu_{\mathcal{B}},\sigma_{\mathcal{B}}}\mathbb{E}_{x\sim p_x}\left[(\mathbb{E}[y|x] - \mathbb{E}_{y|x\sim p_{y|x}}[y])h\right]
$$

$$
= 0.
$$

Since the sample mean converges in probability to the population mean by the Weak Law of Large Numbers, for all $\epsilon > 0$ and a constant number $K$ ($\exists K > 0$ and $\forall P > K$), we have

$p\left(\left|R^f - \mathbb{E}_{(x,y)\sim p_{xy}}\mathbb{E}_{\mu_{\mathcal{B}},\sigma_{\mathcal{B}}}\left[(A'(\bar{h}) - y)h\right]\right| \geq \frac{\rho+2}{4M}\epsilon\right) = 0$. This equation implies that $R^f$ is sufficiently small with a probability of 1 given moderately large number of data points $P$ (the above inequality holds when $P > 30$).

On the other hand, $R^q$ can be derived as

$$
\begin{aligned}
R^q &= \frac{1}{P}\sum_{j=1}^{P}\mathbb{E}_{\mu_{\mathcal{B}},\sigma_{\mathcal{B}}}\left[\frac{A''(\bar{h}^j)}{2}(\hat{h}^j - \bar{h}^j)^2\right]\\
&= \frac{1}{P}\sum_{j=1}^{P}\frac{A''(\bar{h}^j)}{2}\mathbb{E}_{\mu_{\mathcal{B}},\sigma_{\mathcal{B}}}\left[(\gamma\frac{\mathbf{w}^T\mathbf{x}^j - \mu_{\mathcal{B}}}{\sigma_{\mathcal{B}}} + \beta - \gamma\frac{\mathbf{w}^T\mathbf{x}^j - \mu_{\mathcal{P}}}{\sigma_{\mathcal{P}}} + \beta)^2\right]\\
&= \frac{1}{P}\sum_{j=1}^{P}\frac{A''(\bar{h}^j)}{2}\mathbb{E}_{\mu_{\mathcal{B}},\sigma_{\mathcal{B}}}\left[(\gamma\mathbf{w}^T\mathbf{x}^j)^2(\frac{1}{\sigma_{\mathcal{B}}} - \frac{1}{\sigma_{\mathcal{P}}})^2 - 2\gamma\mu_{\mathcal{P}}\mathbf{w}^T\mathbf{x}^j(\frac{1}{\sigma_{\mathcal{B}}} - \frac{1}{\sigma_{\mathcal{P}}})^2 + (\frac{\mu_{\mathcal{B}}}{\sigma_{\mathcal{B}}} - \frac{\mu_{\mathcal{P}}}{\sigma_{\mathcal{P}}})^2\right]\\
&\simeq \frac{1}{P}\sum_{j=1}^{P}\frac{\gamma^2 A''(\bar{h}^j)}{2}\left((\mathbf{w}^T\mathbf{x}^j - \mu_{\mathcal{P}})^2\mathbb{E}_{\mu_{\mathcal{B}},\sigma_{\mathcal{B}}}\left[(\frac{1}{\sigma_{\mathcal{B}}} - \frac{1}{\sigma_{\mathcal{P}}})^2\right] + \mathbb{E}_{\mu_{\mathcal{B}},\sigma_{\mathcal{B}}}\left[\left(\frac{\mu_{\mathcal{B}} - \mu_P}{\sigma_{\mathcal{B}}}\right)^2\right]\right)\\
&= \frac{1}{P}\sum_{j=1}^{P}\frac{\gamma^2 A''(\bar{h}^j)}{2}\left((\frac{\mathbf{w}^T\mathbf{x}^j - \mu_{\mathcal{P}}}{\sigma_{\mathcal{P}}})^2\frac{\rho+2}{4M} + \frac{1}{M}(1 + \frac{3(\rho+2)}{4M})\right).
\end{aligned}
$$

Note that $\frac{\partial^2 l(\bar{h}^j)}{\partial \gamma^2} = A''(\bar{h}^j)(\frac{\mathbf{w}^T\mathbf{x}^j - \mu_{\mathcal{P}}}{\sigma_{\mathcal{P}}})^2$, we have $\mathcal{I}(\gamma) = \frac{1}{P}\sum_{j=1}^{P}A''(\bar{h}^j)(\frac{\mathbf{w}^T\mathbf{x}^j - \mu_{\mathcal{P}}}{\sigma_{\mathcal{P}}})^2$ been an estimator of the Fisher information with respect to the scale parameter $\gamma$. Then, by neglecting $O(1/M^2)$ high-order term in $R^q$, we get

$$
R^q \simeq \frac{\rho+2}{8M}\mathcal{I}(\gamma)\gamma^2 + \frac{\mu_{d^2 A}}{2M}\gamma^2,
$$

where $\mu_{d^2 A}$ indicates the mean of the second derivative of $A(h)$. $\qquad\square$

The results of both ReLU activation function and identity function are provided as below.

## C.2 ReLU Activation Function

For the ReLU non-linear activation function, that is $g(h) = \max(h, 0)$, we use its continuous approximation softplus function $g(h) = \log(1 + \exp(h))$ to derive the partition function $A(h)$. In this case, we have $\mu_{d^2 A} = \frac{1}{P}\sum_{j=1}^{P}\sigma(\bar{h}^j)$. Therefore, we have $\zeta(h) = \frac{\rho+2}{8M}\mathcal{I}(\gamma) + \frac{1}{2M}\frac{1}{P}\sum_{j=1}^{P}\sigma(\bar{h}^j)$ as shown in Eqn.(3).

## C.3 Linear Student Network with Identity Activation Function

For a loss function with identity (linear) units, $\frac{1}{P}\sum_{j=1}^{P}\left(\mathbf{w}^{*\mathsf{T}}\mathbf{x}^j - \gamma(\mathbf{w}^\mathsf{T}\mathbf{x}^j - \mu_{\mathcal{B}})/\sigma_{\mathcal{B}}\right)^2$, we have $\mathcal{I}(\gamma) = 2\lambda$ and $\rho = 0$ for Gaussian input distribution. The exact expression of Eqn.(3) is also possible for such linear regression problem. Under the condition of Gaussian input $\mathbf{x} \sim \mathcal{N}(0, 1/N)$, $h = \mathbf{w}^\mathsf{T}\mathbf{x}$ is also a random variable satisfying a normal distribution $\mathcal{N}(0, 1)$. It can be derived that $\mathbb{E}\left(\sigma_{\mathcal{B}}^{-1}\right) = \frac{\sqrt{M}}{\sqrt{2}\sigma_{\mathcal{P}}}\frac{\Gamma\left(\frac{M-2}{2}\right)}{\Gamma\left(\frac{M-1}{2}\right)}$ and $\mathbb{E}\left(\sigma_{\mathcal{B}}^{-2}\right) = \frac{M}{\sigma_{\mathcal{P}}^2}\frac{\Gamma\left(\frac{M-1}{2}-1\right)}{\Gamma\left(\frac{M-1}{2}\right)}$. Therefore

$$
\zeta = \lambda\left(1 + \frac{M\Gamma\left((M-3)/2\right)}{2\Gamma\left((M-1)/2\right)} - \sqrt{2M}\frac{\Gamma\left((M-2)/2\right)}{\Gamma\left((M-1)/2\right)}\right).
$$

Furthermore, the expression of $\zeta$ can be simplified as $\zeta = \frac{3}{4M}$. If the bias term is neglected in a simple linear regression, contributions from $\mu_{\mathcal{B}}$ to the regularization term is neglected and thus $\zeta = \frac{1}{4M}$. Note that if one uses mean square error without being divided by 2 during linear regression, the values for $\zeta$ should be multiplied by 2 as well, where $\zeta = \frac{1}{2M}$.

## C.4  BN Regularization in a Deep Network

The previous derivation is based on the single-layer perceptron. In deep neural networks, the forward computation inside one basic building block of a deep network is written by

$$z_i^l = g(\hat{h}_i), \quad \hat{h}_i^l = \gamma_i^l \frac{h_i^l - (\mu_{\mathcal{B}})_i^l}{(\sigma_B)_i^l} + \beta_i^l \quad \text{and} \quad h_i^l = (\mathbf{w}_i^l)^{\mathsf{T}} \mathbf{z}^{l-1}, \tag{7}$$

where the superscript $l \in [1, L]$ is the index of a building block in a deep neural network, and $i \in [1, N^l]$ indexes each neuron inside a layer. $z^0$ and $z^L$ are synonyms of input $x$ and output $y$, respectively. In order to analyze the regularization of BN from a specific layer, one needs to isolate its input and focus on the noise introduced by the BN layer in this block. Therefore, the loss function $\ell(\hat{h}^l)$ can also be expanded at $\ell(\bar{h}^l)$. In BN, the batch variance is calculated with regard to each neuron under the assumption of mutual independence of neurons inside a layer. By following this assumption and the above derivation in Appendix C.1, the loss function with BN in deep networks can also be similarly decomposed.

**Regularization of $\mu_{\mathcal{B}}^l, \sigma_{\mathcal{B}}^l$ in a deep network.** Let $\zeta^l$ be the strength (coefficient) of the regularization at the $l$-th layer. Then

$$\frac{1}{P}\sum_{j=1}^{P} \mathbb{E}_{\mu_{\mathcal{B}}^l, \sigma_{\mathcal{B}}^l} \ell\big((\hat{h}^l)^j\big) \simeq \frac{1}{P}\sum_{j=1}^{P} \ell\big((\bar{h}^l)^j\big) + \sum_i^{N^l} \zeta_i^l \cdot (\gamma_i^l)^2,$$

$$\text{and} \quad \zeta_i^l = \frac{1}{P}\sum_{j=1}^{P} \frac{\text{diag}\big(\mathcal{H}_\ell(\bar{h}^l)^j\big)_i}{2} \left( \frac{\rho_i^l + 2}{4M}\left(\frac{(\mathbf{w}_i^l)^T (\mathbf{z}^{l-1})^j - (\mu_{\mathcal{P}})_i^l}{(\sigma_{\mathcal{P}})_i^l}\right)^2 + \frac{1}{M} \right) + \mathcal{O}(1/M^2),$$

where $i$ is the index of a neuron in the layer, $(\bar{h}_i^l)^j = \gamma_i^l \frac{(\mathbf{w}_i^l)^T (\mathbf{z}^{l-1})^j - (\mu_{\mathcal{P}})_i^l}{(\sigma_{\mathcal{P}})_i^l} + \beta_i^l$ represents population normalization (PN), $\mathcal{H}_\ell(\bar{h}^l)$ is the Hessian matrix at $\bar{h}^l$ regarding to the loss $\ell$ and $\text{diag}(\cdot)$ represents the diagonal vector of a matrix.

It is seen that the above equation is compatible with the results from the single-layer perceptron. The main difference of the regularization term in a deep model is that the Hessian matrix is not guaranteed to be positive semi-definite during training. However, this form of regularization is also seen from other regularization such as noise injection (Rifai et al., 2011) and dropout (Wager et al., 2013), and has long been recognized as a Tikhonov regularization term (Bishop, 1995).

In fact, it has been reported that in common neural networks, where convex activation functions such as ReLU and convex loss functions such as common cross entropy are adopted, the Hessian matrix $\mathcal{H}_\ell(\bar{h}^l)$ can be seen as 'locally' positive semidefinite (Santurkar et al., 2018). Especially, as training converges to its mimimum training loss, the Hessian matrix of the loss can be viewed as positive semi-definite and thus the regularization term on $\gamma^l$ is positive.

## C.5  Dynamical Equations

Here we discuss the dynamical equations of BN. Let the length of teacher's weight vector be 1, that is, $\frac{1}{N}\mathbf{w}^{*\mathsf{T}}\mathbf{w}^* = 1$. We introduce a normalized weight vector of the student as $\widetilde{\mathbf{w}} = \sqrt{N}\gamma \frac{\mathbf{w}}{\|\mathbf{w}\|}$. Then the overlapping ratio between teacher and student, the length of student's vector, and the length of student's normalized weight vector are $\frac{1}{N}\widetilde{\mathbf{w}}^{\mathsf{T}}\mathbf{w}^* = QR = \gamma R$, $\frac{1}{N}\widetilde{\mathbf{w}}^{\mathsf{T}}\widetilde{\mathbf{w}} = Q^2 = \gamma^2$, and $\frac{1}{N}\mathbf{w}^{\mathsf{T}}\mathbf{w} = L^2$ respectively, where $Q = \gamma$. And we have $\frac{1}{N}\mathbf{w}^{\mathsf{T}}\mathbf{w} = LR$.

We transform update equations (5) by using order parameters. The update rule for variable $Q^2$ can be obtained by $(Q^2)^{j+1} - (Q^2)^j = \frac{1}{N}\big[2\eta\delta^j\tilde{\mathbf{w}}^{j\mathsf{T}}\mathbf{x}^j - 2\eta\zeta(Q^2)^j\big]$ following update rule of $\gamma$. Similarly, the update rules for variables $RL$ and $L^2$ are calculated as follow:

$$(RL)^{j+1} - (RL)^j = \frac{1}{N}\Big(\frac{\eta Q^j}{L^j}\delta^j\mathbf{w}^{*\mathsf{T}}\mathbf{x}^j - \frac{\eta R^j}{L^j}\delta^j\tilde{\mathbf{w}}^{j\mathsf{T}}\mathbf{x}^j\Big),$$

$$(L^2)^{j+1} - (L^2)^j = \frac{1}{N}\Big[\frac{\eta^2(Q^2)^j}{(L^2)^j}\delta^{j^2}\mathbf{x}^{j\mathsf{T}}\mathbf{x}^j - \frac{\eta^2}{N(L^2)^j}\delta^{j^2}(\tilde{\mathbf{w}}^{j\mathsf{T}}\mathbf{x}^j)^2\Big]. \tag{8}$$

Let $t = \frac{j}{N}$ is a normalized sample index that can be treated as a continuous time variable. We have $\Delta t = \frac{1}{N}$ that approaches zero in the thermodynamic limit when $N \to \infty$. In this way, the learning dynamic of $Q^2$, $RL$ and $L^2$ can be formulated as the following differential equations:

$$\begin{cases} \frac{dQ^2}{dt} & = 2\eta I_1 - 2\eta\zeta Q^2, \\ \frac{dRL}{dt} & = \eta\frac{Q}{L}I_3 - \eta\frac{R}{L}I_1, \\ \frac{dL^2}{dt} & = \eta^2\frac{Q^2}{L^2}I_2, \end{cases} \qquad (9)$$

where $I_1 = \langle\delta\tilde{\mathbf{w}}^\mathsf{T}\mathbf{x}\rangle_\mathbf{x}$, $I_2 = \langle\delta^2\mathbf{x}^\mathsf{T}\mathbf{x}\rangle_\mathbf{x}$, and $I_3 = \langle\delta\mathbf{w}^{*\mathsf{T}}\mathbf{x}\rangle_\mathbf{x}$, which are the terms presented in $\frac{dQ^2}{dt}$, $\frac{dRL}{dt}$, and $\frac{dL^2}{dt}$ and $\langle\cdot\rangle_\mathbf{x}$ denotes expectation over the distribution of $\mathbf{x}$. They are used to simplify notations. Note that we neglect the last term of $dL^2/dt$ in Eqn.(8) since $\frac{\eta^2}{N(L^2)}\delta^2(\tilde{\mathbf{w}}^\mathsf{T}\mathbf{x})^2$ can be approximately equal to zero when $N$ approaches infinity. On the other hand, we have $dQ^2 = 2QdQ, dRL = RdL + LdR$ and $dL^2 = 2LdL$. Hence, Eqn.(9) can be reduced to

$$\begin{cases} \frac{dQ}{dt} & = \eta\frac{I_1}{Q} - \eta\zeta Q, \\ \frac{dR}{dt} & = \eta\frac{Q}{L^2}I_3 - \eta\frac{R}{L^2}I_1 - \eta^2\frac{Q^2R}{2L^4}I_2, \\ \frac{dL}{dt} & = \eta^2\frac{Q^2}{2L^3}I_2. \end{cases} \qquad (10)$$

**Proposition 1.** *Let $(Q_0, R_0, L_0)$ denote a fixed point with parameters $Q$, $R$ and $L$ of Eqn.(10). Assume the learning rate $\eta$ is sufficiently small when training converges and $x \sim \mathcal{N}(0, \frac{1}{N}\mathbf{I})$. If activation function $g$ is* ReLU*, then we have $Q_0 = \frac{1}{2\zeta+1}, R_0 = 1$ and $L_0$ could be arbitrary.*

*Proof.* First, $L$ has no influence on the output of student model since $\mathbf{w}$ is normalized, which implies that if $(Q_0, R_0, L_0)$ is a fixed point of Eqn.(10), $L_0$ could be arbitrary. Besides, we have $\eta \gg \eta^2$ because the learning rate $\eta$ is sufficiently small. Therefore, the terms in Eqn.(10) proportional to $\eta^2$ can be neglected. If $(Q_0, R_0, L_0)$ is a fixed point, it suffices to have

$$\eta\frac{I_1(Q_0, R_0)}{Q_0} - \eta\zeta Q_0 = 0, \qquad (11)$$

$$\eta\frac{Q_0}{L_0^2}I_3(Q_0, R_0) - \eta\frac{R_0}{L_0^2}I_1(Q_0, R_0) = 0, \qquad (12)$$

To calculate $I_1$ and $I_3$, we define $s$ and $t$ as $\tilde{\mathbf{w}}^\mathsf{T}\mathbf{x}$ and $\mathbf{w}^\mathsf{T}\mathbf{x}$. Since $\mathbf{x} \sim \mathcal{N}(0, \frac{1}{N}\mathbf{I})$, we can acquire

$$\begin{bmatrix} s \\ t \end{bmatrix} \sim N\left(\left(\begin{bmatrix} 0 \\ 0 \end{bmatrix}, \begin{bmatrix} Q^2 & QR \\ QR & 1 \end{bmatrix}\right)\right)$$

so probability measure of $[s, t]^\mathsf{T}$ can be written as

$$DsDt = \frac{1}{2\pi Q\sqrt{1-R^2}}exp\left\{-\frac{1}{2}\begin{bmatrix} s \\ t \end{bmatrix}^T \begin{bmatrix} Q^2 & QR \\ QR & 1 \end{bmatrix}^{-1} \begin{bmatrix} s \\ t \end{bmatrix}\right\}$$

Then,

$$\begin{aligned} I_1 &= \left\langle g'(\tilde{\mathbf{w}}^\mathsf{T}\mathbf{x})\left[g(\mathbf{w}^{*T}\mathbf{x}) - g(\tilde{\mathbf{w}}^\mathsf{T}\mathbf{x})\right]\tilde{\mathbf{w}}^\mathsf{T}\mathbf{x}\right\rangle_\mathbf{x} \\ &= \int_{u,v} [g'(s)\,(g(t) - g(s)\,s]DsDt \\ &= \int_0^{+\infty}\int_0^{+\infty} stDsDt - \int_0^{+\infty} s^2\int_{-\infty}^{+\infty} DsDt \\ &= \frac{Q(\pi R + 2\sqrt{1-R^2} + 2R\arcsin(R))}{4\pi} - \frac{Q^2}{2} \end{aligned} \qquad (13)$$

and

$$
\begin{aligned}
I_3 &= \int\limits_{u,v} [g'(s)\,(g(t) - g(s)\,t]DsDt \\
&= \int\limits_{u,v} g'(s)g(t)tDsDt - \int\limits_{u,v} g'(s)g(s)tDsDt \\
&= \int_0^{+\infty} \int_0^{+\infty} t^2 DsDt - \int_0^{+\infty} \int_{-\infty}^{+\infty} stDsDt \\
&= \frac{\pi + 2R\sqrt{1-R^2} + 2\arcsin(R)}{4\pi} - \frac{QR}{2}
\end{aligned}
\tag{14}
$$

By substituting Eqn.(13) and (14) into Eqn.(11) and (12), we get $Q_0 = \frac{1}{2\zeta+1}$ and $R_0 = 1$. $\qquad\square$

**Proposition 2.** *Given conditions in proposition1, let $\lambda_Q^{bn}$, $\lambda_R^{bn}$ be the eigenvalues of the Jacobian matrix at fixed point $(Q_0, R_0, L_0)$ corresponding to the order parameters $Q$ and $R$ respectively in BN. Then*

$$
\begin{cases}
\lambda_Q^{bn} = \frac{\eta}{Q_0} \frac{\partial I_1}{\partial Q} - \eta\zeta Q_0, \\
\lambda_R^{bn} = \frac{\partial I_2}{2\partial R} \frac{\eta Q_0}{2L_0^2}(\eta_{\max}^{bn} - \eta_{\text{eff}}^{bn}),
\end{cases}
$$

*where $\eta_{\max}^{bn}$ and $\eta_{\text{eff}}^{bn}$ are the maximum and effective learning rates respectively in BN.*

*Proof.* At fixed point $(Q_0, R_0, L_0) = (\frac{1}{2\zeta+1}, 1, L_0)$ obtained in proposition1, the Jacobian of dynamic equations of BN can be derived as

$$
J^{bn} = \begin{bmatrix}
\frac{\eta}{Q_0}\frac{\partial I_1}{\partial Q} - 2\eta\zeta & \frac{\eta}{Q_0}\frac{\partial I_1}{\partial R} & 0 \\
0 & \frac{\eta}{L_0^2}\left(\frac{Q_0 \partial I_3}{\partial R} - \frac{\partial I_1}{\partial R} - \zeta Q_0^2\right) - \frac{\eta^2 Q_0^2}{2L_0^4}\frac{\partial I_2}{\partial R} & 0 \\
0 & \frac{\eta^2 Q_0^2}{2L_0^3}\frac{\partial I_2}{\partial R} & 0
\end{bmatrix},
$$

and the eigenvalues of $J^{bn}$ can be obtained by inspection

$$
\begin{cases}
\lambda_Q^{bn} = \frac{\eta}{Q_0}\frac{\partial I_1}{\partial Q} - 2\eta\zeta, \\
\lambda_R^{bn} = \frac{\eta}{L_0^2}\left(\frac{Q_0 \partial I_3}{\partial R} - \frac{\partial I_1}{\partial R} - \zeta Q_0^2\right) - \frac{\eta^2 Q_0^2}{2L_0^4}\frac{\partial I_2}{\partial R} = \frac{\partial I_2}{\partial R}\frac{\eta Q_0}{2L_0^2}\left(\eta_{\max}^{bn} - \eta_{\text{eff}}^{bn}\right), \\
\lambda_L^{bn} = 0.
\end{cases}
$$

Since $\gamma_0 = Q_0$, we have $\eta_{\max}^{bn} = \left(\frac{\partial(\gamma_0 I_3 - I_1)}{\gamma_0 \partial R} - \zeta\gamma_0\right)/\frac{\partial I_2}{2\partial R}$ and $\eta_{\text{eff}}^{bn} = \frac{\eta\gamma_0}{L_0^2}$. $\qquad\square$

## C.6  STABLE FIXED POINTS OF BN

**Proposition 3.** *Given conditions in proposition1, when activation function is ReLU, then (i) $\lambda_Q^{bn} < 0$, and (ii) $\lambda_R^{bn} < 0$ iff $\eta_{\max}^{bn} > \eta_{\text{eff}}^{bn}$.*

*Proof.* When activation function is ReLU, we derive $I_1 = \frac{Q(\pi R + 2\sqrt{1-R^2} + 2R\arcsin(R))}{4\pi} - \frac{Q^2}{2}$, which gives

$$
\frac{\partial I_1}{\partial Q} = -Q + \frac{\pi R + 2\sqrt{1-R^2} + 2R\arcsin(R)}{4\pi}.
$$

Therefore at the fixed point of BN $(Q_0, R_0, L_0) = (\frac{1}{2\zeta+1}, 1, L_0)$, we have

$$
\lambda_Q^{bn} = \eta(\frac{1}{Q_0}\frac{\partial I_1}{\partial Q} - 2\zeta) = \eta(\frac{1}{Q_0}(-1 + \frac{1}{2Q_0} - 2\zeta) = -\zeta - \frac{1}{2} < 0.
$$

Note that $\mathbf{x}^\mathsf{T}\mathbf{x}$ approximately equals 1. We get

$$
\begin{aligned}
I_2 &= \int_{u,v} [g'(s)\,(g(t)-g(s))]^2 DsDt \\
&= \int_0^{+\infty}\int_0^{+\infty} v^2 DsDt + \int_0^{+\infty}\int_{-\infty}^{+\infty} s^2 DsDv - 2\int_0^{+\infty}\int_{-\infty}^{+\infty} stDsDt \\
&= \frac{Q^2}{2} + \frac{\pi R + 2R\sqrt{1-R^2} + 2\arcsin(R)}{4\pi} - \frac{Q(\pi R + 2\sqrt{1-R^2}+2R\arcsin(R))}{2\pi}.
\end{aligned}
\tag{15}
$$

At the fixed point we have $\frac{\partial I_2}{\partial R} = -Q_0 < 0$. Therefore, we conclude that $\lambda_R^{bn} < 0$ iff $\eta_{\max}^{bn} > \eta_{\text{eff}}^{bn}$. $\qquad\square$

## C.7 Maximum Learning Rate of BN

**Proposition 4.** *When the activation function is ReLU, then $\eta_{\max}^{bn} \geq \eta_{\max}^{\{wn,sgd\}} + 2\zeta$, where $\eta_{\max}^{bn}$ and $\eta_{\max}^{\{wn,sgd\}}$ indicate the maximum learning rates of BN, WN, and vanilla SGD respectively.*

*Proof.* From the above results, we have $I_1 = \frac{Q(\pi R + 2\sqrt{1-R^2}+2R\arcsin(R))}{4\pi} - \frac{Q^2}{2}$, which gives $\partial I_1/\partial R \geq 0$ at the fixed point of BN. Then it can be derived that $\frac{\partial I_2}{\partial R} < 0$. Furthermore, at the fixed point of BN, $Q_0 = \gamma_0 = \frac{1}{2\zeta+1} < 1$, then we have

$$
\begin{aligned}
\eta_{\max}^{bn} &= \left(\frac{\partial(\gamma_0 I_3 - I_1)}{\gamma_0 \partial R} - \zeta\gamma_0\right)\Big/\frac{\partial I_2}{2\partial R} \\
&= \frac{\partial(I_3-I_1)}{\partial R}\Big/\frac{\partial I_2}{2\partial R} + (1-\frac{1}{\gamma_0})\frac{\partial I_1}{\partial R}\Big/\frac{\partial I_2}{2\partial R} - \zeta\gamma_0\Big/\frac{\partial I_2}{2\partial R} \\
&\geq \frac{\partial(I_3-I_1)}{\partial R}\Big/\frac{\partial I_2}{2\partial R} + 2\zeta
\end{aligned}
$$

where the inequality sign holds because $(1-\frac{1}{\gamma_0})\frac{\partial I_1}{\partial R}\big/\frac{\partial I_2}{2\partial R}$ is positive. Note that $\frac{\partial(I_3-I_1)}{\partial R}\big/\frac{\partial I_2}{2\partial R}$ is also defined as maximum learning rates of WN, and vanilla SGD in Yoshida et al. (2017). Hence, we conclude that $\eta_{\max}^{bn} \geq \eta_{\max}^{\{wn,sgd\}} + 2\zeta$. $\qquad\square$

## D Proofs regarding generalization and statistical mechanics (SM)

In this section, we build an analytical model for the generalization ability of a single-layer network. The framework is based on the Teacher-Student model, where the teacher network output $y^* = g^*\left(\mathbf{w}^{*\mathsf{T}}\cdot\mathbf{x}+s\right)$ is learned by a student network. The weight parameter of the teacher network satisfies $\frac{1}{N}(\mathbf{w}^*)^\mathsf{T}\cdot\mathbf{w}^*=1$ and the bias term $s$ is a random variable $s \sim \mathcal{N}(0,S)$ fixed for each training example $\mathbf{x}$ to represent static errors in training data from observations. In the generalization analysis, the input is assumed to be drawn from $\mathbf{x} \sim \mathcal{N}\left(0,\frac{1}{N}\mathbf{I}\right)$. The output of the student can also be written as a similar form $y = g(\widetilde{\mathbf{w}}\cdot\mathbf{x})$, where the activation function $g(\cdot)$ can be either linear or ReLU in the analysis and $\widetilde{\mathbf{w}}$ is a general weight parameter which can be used in either WN or common linear perceptrons. Here we take WN for example, since it has been derived in this study that BN can be decomposed into WN with a regularization term on $\gamma$. In WN $\widetilde{\mathbf{w}} = \gamma\frac{\mathbf{w}}{\|\mathbf{w}\|_2}$ and we defined the same order parameter as the previous section that $\gamma^2 = \frac{1}{N}\widetilde{\mathbf{w}}^\mathsf{T}\cdot\widetilde{\mathbf{w}}$ and $\gamma R = \frac{1}{N}\widetilde{\mathbf{w}}^\mathsf{T}\cdot\mathbf{w}^*$.

### D.1 Generalization error

Since the learning task is a regression problem with teacher output biased by a Gaussian noise, it comes natural that we can use the the average mean square error loss $\epsilon_t = \frac{1}{P}\sum_j \left(y_j^* - y_j\right)^2$ for the regression. The generalization error defined as the estimation over the distribution of input $\mathbf{x}$ and is written as

$$
\epsilon_{\text{gen}}(\widetilde{\mathbf{w}}) = \left\langle (y^* - y)^2 \right\rangle_{\mathbf{x}}
\tag{16}
$$

where $\langle \cdot \rangle_{\mathbf{x}}$ denotes an average over the distribution over $\mathbf{x}$. The generalization error is a function of its weight parameter and can be converted to a function only with regard to the aforementioned order parameters, detailed derivation can be seen in (Bös, 1998; Krogh & Hertz, 1992).

$$\epsilon_{\text{gen}}(\gamma, R) = \iint Dh_1 Dh_2 \left[ g^*(h_1) - g(\gamma R h_1 + \gamma \sqrt{1-R^2} h_2) \right]^2 \tag{17}$$

where $h_1$ and $h_2$ are variables drawn from standard Gaussian distribution and $Dh_1 := \mathcal{N}(0,1) dh_1$.

When both the teacher network and student network have a linear activation function, the above integration can be easily solved and

$$\epsilon_{\text{gen}}(\gamma, R) = 1 + \gamma^2 - 2\gamma R \tag{18}$$

As for the case where the teacher network is linear and the student network has a ReLU activation, it can still be solved first by decomposing the loss function

$$
\begin{aligned}
\epsilon_{\text{gen}}(\gamma, R) &= \iint Dh_1 Dh_2 \left[ h_1 - g(\gamma R h_1 + \gamma \sqrt{1-R^2} h_2) \right]^2 \\
&= \iint Dh_1 Dh_2 \left[ h_1^2 + g(\gamma R h_1 + \gamma \sqrt{1-R^2} h_2)^2 - 2h_1 g(\gamma R h_1 + \gamma \sqrt{1-R^2} h_2) \right]^2 \\
&= 1 + \frac{\gamma^2}{2} - 2 \iint Dh_1 Dh_2 \left[ h_1 g(\gamma R h_1 + \gamma \sqrt{1-R^2} h_2) \right]^2
\end{aligned}
$$

It should be noted that the last two terms should only be integrated over the half space $\gamma R h_1 + \gamma \sqrt{1-R^2} h_2 > 0$, and therefore if we define the angle of this line with the $h_2$ axis $\theta_0 = \arccos(R)$ the integration is transformed to polar coordinate

$$
\begin{aligned}
\epsilon_{\text{gen}}(\gamma, R) &= 1 + \frac{\gamma^2}{2} - 2 \iint Dh_1 Dh_2 \left[ h_1 g(\gamma R h_1 + \gamma \sqrt{1-R^2} h_2) \right]^2 \\
&= 1 + \frac{\gamma^2}{2} - 2 \int_{-\theta_0}^{\pi-\theta_0} d\theta \int_0^\infty r dr \frac{1}{2\pi} \exp(-\frac{r^2}{2}) \left( \gamma R r^2 \sin^2(\theta) + \gamma \sqrt{1-R^2} r^2 \cos(\theta) \sin(\theta) \right) \\
&= 1 + \frac{\gamma^2}{2} - \gamma R
\end{aligned}
$$

### D.2 Equilibrium order parameters

Following studies on statistical mechanics, the learning process of a neural network resembles a Langevin process (Mandt et al., 2017) and at the equilibrium the network parameters $\theta$ follow a Gibbs distribution. That is, the weight vector that yields lower training error produces higher probability. We have $p(\theta) = Z^{-1} \exp\{-\beta \epsilon_t(\theta; \mathbf{x})\}$, where $\beta = 1/T$ and $T$ is temperature, representing the variance of noise during training and implicitly controlling the learning process. $\epsilon_t(\theta; \mathbf{x})$ is an energy term of the training loss function, $Z = \int d\mathcal{P}(\theta) \exp\{-\epsilon_t(\theta; \mathbf{x})/T\}$ is the partition function, and $\mathcal{P}(\theta)$ is a prior distribution.

Instead of directly minimizing the energy term above, statistical mechanics finds the minima of free energy, $f$, which is a function over $T$, considering the fluctuations of $\theta$ at finite temperatures. We have $-\beta f = \langle \ln Z \rangle_{\mathbf{x}}$.

By substituting the parameters that minimize $f$ back into the generalization errors calculated above, we are able to calculate the averaged generalization error, at a certain temperature.

The solution of SM requires the differentiation of $f$ with respect to the order parameters.

In general, the expression of free energy under the replica theory is expressed as(Seung et al., 1992)

$$-\beta f = \frac{1}{2}\frac{(\gamma^2 - \gamma^2 R^2)}{q^2 - \gamma^2} + \frac{1}{2}\ln(q^2 - \gamma^2) + \alpha \iint Dh_1 Dh_2 \ln \left[\int Dh_3 \exp\left(-\frac{\beta(g-g^*)^2}{2}\right)\right]$$

$$\tag{19}$$

where

$$g := g\left(\gamma R h_1 + \sqrt{\gamma^2 - \gamma^2 R^2}h_2 + \sqrt{q^2 - \gamma^2}h_3\right)$$

$$g^* := g^*(h_1 + s)$$

$$\tag{20}$$

In the above expression, $h_1, h_2, h_3$ are three independent variables following the standard Gaussian distribution and $\alpha = P/N$ represents the ratio of the number of training samples $P$ to number of unknown parameters $N$ in the network, $R = \frac{1}{N}\frac{\mathbf{w}}{\|\mathbf{w}\|_2} \cdot \mathbf{w}^*$, $q$ is the prior value of $\gamma$..

The above equation can be utilized for a general SM solution of a network. However, the solution is notoriously difficult to solve and only a few linear settings for the student network have close-form solutions(Bös, 1998). Here we extend the previous analysis of linear activations to a non-linear one, though still under the condition that $\beta \to \infty$, which means that the student network undergoes a exhaustive learning that minimizes the training error. In the current setting, the student network is a nonlinear ReLU network while the teacher is a noise-corrupted linear one.

**Proposition 5.** *Given a single-layer linear teacher $y^* = \mathbf{w}^*\mathbf{x} + s$ and a student ReLU network $y = g(\gamma\frac{\mathbf{w}}{\|\mathbf{w}\|_2}\mathbf{x})$ linear student network with g being a ReLU activation function, $\mathbf{x} \sim \mathcal{N}(0, \frac{\mathbf{I}}{N})$ the free energy $f$ satisfies as $\beta \to \infty$*

$$-\beta f = \frac{1}{2}\frac{(\gamma^2 - \gamma^2 R^2)}{q^2 - \gamma^2} + \frac{1}{2}\ln(q^2 - \gamma^2)$$

$$- \frac{\alpha}{4}\ln\left(1 + \beta\left(q^2 - \gamma^2\right)\right) - \frac{\alpha\beta\left(1 - 2\gamma R + \gamma^2 + S\right)}{4\left(1 + \beta\left(q^2 - \gamma^2\right)\right)} - \frac{\alpha\beta}{4} - \frac{\alpha\beta}{4}S$$

$$\tag{21}$$

*where S is the variance of the Gaussian noise s injected to the output of the teacher.*

*Proof.* The most difficult process in Eqn.19 is to solve the inner integration over $h_3$. Here as $\beta \to \infty$, it is noted that the function $\exp\left(-\beta x\right)$ only notches up only at $x = 0$ and is 0 elsewhere. Therefore, the integration $\int Dh_3 \exp\left(-\frac{\beta(g-g^*)^2}{2}\right)$ depends on the value of $g^*$. If $g^* < 0$, no solution exists for $g - g^* = 0$ as $g$ is a ReLU activation, and thus the integration is equivalent to the maximum value of the integral under the limit of $\beta \to \infty$. As $g^* > 0$, the integration over the "notch" is equivalent to the one at full range. That is,

$$\int Dh_3 \exp\left(-\frac{\beta(g-g^*)^2}{2}\right) = \begin{cases} \int Dh_3 \exp\left(-\frac{\beta(g-g^*)^2}{2}\right) & h_1 + s > 0 \\ \max_{h_3} \exp\left(-\frac{\beta(g-g^*)^2}{2}\right) & h_1 + s \le 0 \end{cases}$$

The above equation can be readity integrated out and we obtain

$$\ln\int Dh_3 \exp\left(-\frac{\beta(g-g^*)^2}{2}\right) = -\frac{1}{2}\ln\left(1 + \beta\left(q^2 - \gamma^2\right)\right)$$

$$- \frac{\beta}{2}\frac{\left((1 - \gamma R)h_1 - \sqrt{\gamma^2 - \gamma^2 R^2}h_2 + s\right)^2}{1 + \beta\left(q^2 - \gamma^2\right)}$$

Substituting it back to Eqn.19, we have its third term equivalent

$$\text{Term3} = \alpha \iint_{h_1+s>0} Dh_1 Dh_2 \left[ -\frac{\beta}{2} \frac{\left( (1-\gamma R) h_1 - \sqrt{\gamma^2 - \gamma^2 R^2} h_2 + s \right)^2}{1 + \beta (q^2 - \gamma^2)} \right]$$

$$= \alpha \iint_{h_1+s>0} Dh_1 Dh_2 \left[ -\frac{\beta}{2} \frac{\left( (1-\gamma R)^2 h_1^2 - \sqrt{\gamma^2 - \gamma^2 R^2} h_2 + s \right)^2}{1 + \beta (q^2 - \gamma^2)} \right]$$

To solve the above integration, we first realize that $s$ is a random variable to corrupt the output of the teacher output and the above integration should be averaged out over $s$. Given that $s \sim \mathcal{N}(0, S)$, it is easy to realize that

$$\left\langle \int_{h+s>0} s^2 Dh \right\rangle_s = \frac{S}{2}, \quad \left\langle \int_{h+s>0} Dh \right\rangle_s = \frac{1}{2}, \text{ and } \left\langle \int_{h+s>0} hs Dh \right\rangle_s = 0$$

Through simple Gaussian integraions, we get

$$\text{Term3} = \alpha \left[ -\frac{1}{4} \ln \left( 1 + \beta (q^2 - \gamma^2) \right) - \frac{\beta (1 - 2\gamma R + \gamma^2 + S)}{4 (1 + \beta (q^2 - \gamma^2))} - \frac{\beta}{4} - \frac{\beta}{4} S \right]$$

Substituting Term3 back yields the results of the free energy. $\qquad \square$

Therefore, by locating the values that minimizes $f$ in the above proposition, we have equilibrium order parameters

$$\gamma^2 = \frac{\alpha}{2a} + \frac{\alpha S}{2a - \alpha} \tag{22}$$

and

$$\gamma R = \frac{\alpha}{2a} \tag{23}$$

where $a$ is defined as $a = \frac{1 + \beta(q^2 - \gamma^2)}{\beta(q^2 - \gamma^2)}$. Substituting the order parameters back to the generalization error, we have

$$\epsilon_{\text{gen}} = 1 - \frac{\alpha}{4a} + \frac{\alpha S}{2a (2a - \alpha)} \tag{24}$$

When $\alpha < 2$ and $\beta \to \infty$, $a = 1$, the generalization error is

$$\epsilon_{\text{gen}} = 1 - \frac{\alpha}{4} + \frac{\alpha S}{2 (2 - \alpha)} \tag{25}$$

