# OpenReview forum: "Towards Understanding Regularization in Batch Normalization"
_ICLR.cc/2019/Conference_

### Official Review · AnonReviewer3 · 2018-10-31
**Too much (interesting) content in too little space. Found it hard to follow.**

**Rating:** 6
**Confidence:** 2

**Review:**

This paper investigates batch normalization from three points of view. i) Loss decomposition, ii) learning rate selection, iii) generalization. If carefully read, I believe authors have interesting results and insightful messages. However, as a whole, I found the paper difficult to follow. Too much content is packed into too little space and they are not necessarily coherent with each other. Many of the technical terms are not motivated and even not defined. Overall, cleaning up the exposition would help a lot for readability.

I have a few other technical comments.
1) Theorem 1 is not acceptable for publication. It is not a rigorous statement. This should be fixed.
2) Effective and maximum learning rate is not clear from the main body of the paper. I can intuitively guess what they are but they lack motivation and definition (as far as I see).
3) In Section 3 I believe random data is being assumed (there is expectation over x in some notation). This should be stated upfront. Authors should broadly comment on the applicability of the learning rates calculated as N->\infty in the finite N,P regime?

---

> ### Author Response · Authors · 2018-11-26
> **Response to AnonReviewer3**
>
> We thank the review for the patient reviewing and helpful advice.
>
> As you have pointed out, we mainly contributed to the community in three parts: loss decomposition, learning rate selection and generalization of BN. The content may be a little dense, but they are inter-connected to each other. The latter two, namely learning rate selection and generalization analysis are all based on the loss decomposition. And these phenomena have been considered to be windfalls of BN and partially conclusive only from experiments. Therefore, a theoretical and systematic analysis on the effect of BN is necessary in this study.
>
> We also thank the reviewer for the advice to clean up the manuscript to improve its coherence and we are working on it accordingly. Here is the detailed response for the technical comments.
>
> 1) “Theorem 1 is not acceptable for publication. It is not a rigorous statement. This should be fixed”.
> Since we have given proper assumptions of the theorem, we suppose that the reviewer’s concern on Theorem 1 is the same as R1 and R2. The construction of the loss function is firstly based on a single-layer network. Its extension to deep neural networks is also possible, as answered in Appendix C.4 in our latest version of manuscript. Moreover, the regularization effect in deep CNNs is also verified in this paper. We have revised the statement to make it clearer to the reader in the latest version.
>
> 2) In this study, effective learning rate is defined as lr * \gamma_0 / L_0, where \gamma_0 represents the scale parameter at equilibrium position and L_0 is the square average of weight (w) parameters.
>
> Maximum learning rate is the maximum effective lr that allows training to converge without diverging, which maintains the stability of the neural network at a fix point. These two definitions simplify the analysis of learning dynamics but may cause confusion. It has been clarified more in the newest revision.
>
> 3) For the derivation and application of the main theorem, which is the loss decomposition with BN, we did not impose any distribution of the input x. Following this result, the learning dynamics is analyzed in Sec. 3, a general expression of ODEs has been presented without knowing the input distribution.
>
> As for the effective and maximum learning rate at the fixed point,  Gaussian input gives intuitive and meaningful expressions and is thus presented. This has been stated in the latest main text.
>
> 4) The thermodynamic limit (N,P->infinity) allows the asymptotic analysis of learning dynamics from continuous differential equations. In the regime of finite N & P is finite, the differential equation would be replaced by a difference equation. In non-asymptotic regime, higher differential orders of parameters must be accounted for, and the interactions between parameters are more complex [1], and this would normally make the optimization process less stable. In reality, the networks are normally constructed with large number of neurons (~10K in each layer) and data points (~1M), the asymptotic analysis would hold.
>
> [1] E. Moulines and F. R. Bach, “Non-Asymptotic Analysis of Stochastic Approximation Algorithms for Machine Learning,” in Advances in Neural Information Processing Systems 24, 2011, pp. 451–459.

---

### Official Review · AnonReviewer2 · 2018-11-04
**a thought provoking paper**

**Rating:** 6
**Confidence:** 5

**Review:**

This is a thought provoking paper that aims to understand the regularization effects of batch-normalization (BN) under a probabilistic interpretation. The authors connect BN to population normalization (PN) and a gamma-decay term that penalizes the scale of the weights. They analyze the generalization error of BN for a single-layer perceptron using ideas in statistical physics.

Detailed comments:

1. Theorem 1 uses the loss function of a single-layer perceptron in the proof. This is not mentioned in the main writeup. This theorem is not valid in general.

2. The main contribution of this paper is Theorem 1 which connects BN to population normalization and weight normalization. It shows that the regularization of BN can be split into two components that depend on the mini-batch mean and variances: the former penalizes the magnitude of activations while the latter penalizes their correlation.

3. Although the theoretical analysis is conducted under simplistic models, this paper corroborates a number of widely-known observations about BN in practice. It validates these predictions on standard experiments.

4. The scaling of BN regularization with batch-size can be easily seen from Teye et al., 2018, so I think the experiments that validate this prediction are not strictly necessary.

5. It is difficult to use these techniques for deep non-linear networks.

6. The predictions in Section 3.3 are very interesting: it is often seen that fully-connected layers (where BN helps significantly) need small learning rates to train without BN; with BN one can use larger learning rates.

7. The experimental section is very rough. In particular the experiments on CIFAR-10 and downsampled-ImageNet with CNNs seem to have very high errors and it is difficult to understand whether some of the predictions about generalization error apply here. Why not use a more recent architecture for CIFAR-10?

8. There is a very large number of grammatical and linguistic errors in the narrative.

9. The presentation of the paper is very dense, I would advise the authors to move certain parts to the appendix and remove the inlining of important equations to improve readability.

---

> ### Author Response · Authors · 2018-11-26
> **Response to AnonReviewer2**
>
> We thank AnonReviewer2 for the constructive comments. We appreciate the comments on improving the clarity of theorem statements and experimental validations, and the revision of the manuscript is being made accordingly. Besides, we would also like to address several of the above concerns.
>
> The following answers are corresponding to the number (index) of your comments.
>
> 1. As to the assumptions in deriving Theorem 1, it has been stated in the abstract that “we analyze BN by using a basic block of neural networks, consisting of a kernel layer, a BN layer, and a nonlinear activation function”. We have revised the presentation of our results in section 2.1 of the latest manuscript and made this clear.
>
> We modeled the loss function through a probabilistic perspective for a single-layer perceptron. This is treated as an illustrative case, in order to make our discussions as intuitive and easy to understand as possible. The analyses of a single-layer network already explain optimization and generalization of BN compared to the other approaches. Despite the loss construction based on a single layer, we have also verified the major conclusions in deep CNNs.
>
> As for multiple-layer neural networks, the current analysis can be naturally extended. (see Appendix C.4 in the new version of the manuscript).
>
> 4. In the experiment section, the `batch size’ subsection investigates how the strength of regularization of BN affects the parameter norm. This is one of our findings of gamma decay. We’ll consider move this part to Appendix.
>
> 5. The results of section 2.1 can be extended to deep networks. We achieve this by performing decomposition in a deep network with respect to a certain hidden BN layer. The discussions can be found in Appendix C.4. The decay factor of the regularization depends on Hessian matrix of the hidden layer, whose regularization form is not as intuitive and easy to read as the singe-layer ReLU network. We evaluated BN’s regularization in deep CNNs in experiments.
>
> 7. We have revised the experiment section by conducting a strong baseline in CIFAR10 using ResNet18 to study the regularization of BN. We would like to argue that down-sampled ImageNet with image size of 32*32 is more challenging than the full size ImageNet (224*224). That’s why the performance of ResNet18 in the down-sampled version is not comparable to the original ImageNet.
>
> It should be also noted that the purpose of our experiments is to validate the regularization of BN in deep neural networks, instead of improving the performance of networks already pursued with a lot of finely tuned methods. Therefore, in order to focus on the regularization effect of BN, we removed augmentation in data preparation as well as weight decay or dropout to rule out the regularization from these techniques. In this setting in the current paper, the inverse relationship between the strength of BN regularization and batch size was observed most evidently.
>
> 8. We have revised the grammatical errors in the latest manuscript.
>
> 9. As to the presentation of the current paper, thanks for the advice and we have revised it in the latest version of the manuscript.

---

### Official Review · AnonReviewer1 · 2018-11-11

**Rating:** 5
**Confidence:** 3

**Review:**

This is an interesting paper on a statistical analysis of batch normalization. It takes a holistic approach,
combining techniques and ideas from various fields, and considers multiple endpoints, such as tuning of learning rates and estimation of generalization error. Overall it is an interesting paper.

Some aspects of the paper that could be improved:

1) Theorem 1 is not particularly compelling, and may be misleading at a first reading. It considers the simple model of Equation (1) in a straightforward bias-variance decomposition, and may not be useful in general. Some aspects of the theorem are not technically correct or unclear. E.g., \gamma is a single parameter, what does it mean to have a Fisher information matrix?

2) The problem is not motivated well. It may be a good idea to bring some discussions from Section 6 early in the introduction of the paper. When does BN work well? And what is the current understanding (prior to the paper) and how does the paper compare/contribute? I think the paper does a good job on that front, but it follows a disordered narration flow which makes it hard to read. I understand there is a lot of material to cover, but it would help a lot to reorganize the paper in a more linear way.

3) What about alternatives, such as implicit back propagation that stabilizes learning?  [1]

4) I don't find Figure 1 (and 3) particularly useful on how it handles vanilla SGD. In practice, it would be straightforward to avoid the mentioned pathologies. Overall, the experiments are interesting but it may be hard to generalize the findings to non-linear settings.


[1] Implicit back propagation, Fagan & Iyengar, 2017

---

> ### Author Response · Authors · 2018-11-26
> **Response to  AnonReviewer1**
>
> Dear AnnoReviewer1,
>
> Thanks for the constructive comments. We are glad that the reviewer agrees on the necessity and impact of the current work. We list detailed responses to some concerns.
>
> (1) Theorem 1. As we have stated in the main text, we derived the loss decomposition from a single building block in neural networks. We have rephrased section 2.1 to make this clearer. In fact, we started from a single-layer perceptron in order to keep our discussions intuitive and easy to understand.
>
> By analyzing the regularization of BN in a single-layer network, its optimization and generalization are investigated sufficiently and BN is compared to WN+gamma decay and vanilla SGD both theoretically and numerically, as shown in section 3 and 4. These results have never been presented before. We believe they should be presented to the community.
>
> In the latest version of manuscript, we extend the regularization form of BN to deep networks as shown in Appendix C.4. We also analyze generalization of BN in a nonlinear ReLU student network in section 4. Moreover, the current study has verified the major findings both analytically in a single-layer network and empirically in deep nonlinear CNNs.
>
> (2) Motivation and narration. We are grateful for this suggestion. We have made modifications in the latest manuscript. For example, the “notation” subsection is removed from the introduction and moved to section 2, the “section 6 Related Work” is moved forward in the introduction to better motivate the problem.
>
>
> (3) Implicit Back Propagation [1]. Thanks for the nice advice to compare with other method that stabilizes training and learning rate. We have cited this reference. However, despite the similarity between BN and implicit BP (ISGD) in the effects of stabilization and robust learning rates, these two methods are quite different.
>
> Implicit BP (ISGD) achieves the above effects by implicitly accounting for higher-order derivatives on the loss surface in backward gradient propagation, while BN reaches this goal by reshaping the loss surface to be more isotropic through normalization in forward computation. From their different perspectives, it seems possible to combine ISGD with BN, but the analysis is beyond the topic of the current study.
>
> (4) Figure 1 (and 3). We agree with the reviewer that vanilla SGD would not be adopted because of its simplest form.
>
> However, the purpose of the comparison in section 4 between vanilla SGD, WN+gamma decay, and BN is to quantitatively verify the findings of BN’s regularization form. In a linear single-layer network, the analytical solutions are easy to obtain and easy to understand for readers.
>
> The extension to non-linear solutions is a bit less straightforward. Thanks for the advice and we have also verified the results in a ReLU nonlinear network as shown in section 4 in the latest manuscript, and made corresponding derivations that are enclosed in Appendix D2.
>
> Hope the above feedback persuade you to raise your confidence and score.

---

### Author Response · Authors · 2018-11-26
**List of changes in the latest manuscript**

List of changes:
1.	The conditions for the regularization form of BN are made clearer in Sec. 2.1.

2.	The extension of BN regularization in deep neural networks has been added in the last paragraph in Sec. 2 and Appendix C.4.

3.	Analytical comparisons of the generalization errors of BN, WN+gamma decay, and vanilla SGD with both identity and ReLU units are included. Moreover, both theoretical and numerical validations have been conducted in Sec. 4.1 and figure 1.

4.	Derivation of the generalization error of a student network with ReLU units under the statistical mechanics theory.  (Appendix D.2)

5.	The network for PN+gamma decay has been changed to a ResNet18 network, which has a much higher baseline as shown in Sec.5.1 Fig.2(a)&(b).

6.	The network for WN+dropout has also been changed to a ResNet18 network, which has a much higher baseline. (Sec.5.1 Fig.2(g)&(h))

7.     The experiment section has been re-organized.

8.	The introduction part has been re-organized by putting forward the “Related Wrok” part and presenting the relationship between the current work and previous ones.

9.	The whole manuscript has been thoroughly scrutinized and proof-read to make it neater and clearer.

---

### Meta-Review · Area_Chair1 · 2018-12-15
**An interesting contribution (that requires more polished exposition)**

**Confidence:** 3
**Recommendation:** Accept (Poster)

**Metareview:**

+ the ideas presented in the paper are quite intriguing and draw on a variety of different connections
- the presentation has a lot of room for improvement. In particular, the statement of Theorem 1, in its current form, requires rephrasing and making it more rigorous.

Still, the general consensus is that, once these presentation shortcomings are address, this will be an interesting paper.